# STOCHASTIC TRUNCATION
# FOR MULTI-STEP OFF-POLICY RL

## ABSTRACT

Multi-step off-policy reinforcement learning is essential for reliable policy evaluation, particularly in long-horizon settings, yet extending beyond one-step temporal-difference learning remains difficult due to distribution mismatch between behavior and target policies. This mismatch is further exacerbated at longer horizons, leading to compounding bias and variance. Existing approaches fall into two categories: *conservative* methods (e.g., Retrace), which guarantee convergence but often suffer from high variance, and *non-conservative* methods (e.g., Peng's $Q(\lambda)$ and integrated algorithms like Rainbow), which often achieve strong empirical performance but do not guarantee convergence under all exploration schemes. We identify horizon selection as the central obstacle and relate it to the mixing time of policy-induced Markov chains. Because mixing time is difficult to estimate online, we derive a practical upper bound via a coupling-based analysis to guide adaptive truncation. Building on this insight, we propose T4[1] (Time To Truncate Trajectory), a stochastic and adaptive truncation mechanism within the Retrace framework. We prove that T4 is non-conservative yet converges under arbitrary behavior policies, and is robust to cap length tuning. T4 improves policy evaluation and control performance over strong baselines on standard RL benchmarks.

## 1 INTRODUCTION

Reinforcement learning (RL) fundamentally relies on *policy evaluation*—the competence to accurately estimate the long-term impact of a policy on future rewards. Accurate policy evaluation is crucial for consistent learning progress and effective decision-making, particularly in long-horizon environments. Multi-step temporal-difference (TD) learning (Mahmood et al., 2017; Asis & Sutton, 2018; Harutyunyan, 2018; Sutton et al., 1998; Precup et al., 2001) leverages long-horizon trajectory information by constructing truncated $n$-step returns, in which the tail is bootstrapped from $Q$-values at the truncation horizon. However, in off-policy RL, the training data are collected by behavior policies whose distributions differ from the evolving target policy. This distribution mismatch inflates the estimation error of the target policy's action-value function, $Q^\pi$, as the truncation horizon $n$ grows, leading to compounding bias and variance. This raises a central question:

*Can multi-step off-policy RL achieve reliable and convergent policy evaluation while effectively mitigating distribution mismatch?*

Prior methods have attempted to address this distribution-mismatch challenge by applying per-step importance weighting to update the $Q$-function toward its Bellman fixed point (Precup et al., 2001; Geist et al., 2014; Farajtabar et al., 2018). Kozuno et al. (2021) classify multi-step off-policy evaluation methods into *conservative* and *non-conservative* categories. Conservative methods ensure convergence under arbitrary behavior policies by modifying the policy evaluation operators, but often incur high variance and instability due to correction ratios that can be excessively large or vanishingly small Rowland et al. (2020). Non-conservative methods relax per-step corrections and often lack general convergence guarantees or rely on restrictive assumptions on the behavior policy.

Estimating reliable weights from policy distributions remains challenging, especially as horizons grow, which hinders the effective extension of one-step off-policy RL to multi-step settings. We

---

[1] Code available at https://anonymous.4open.science/r/t4-BD20

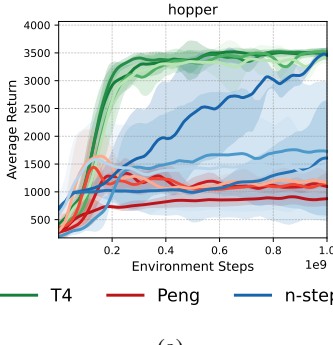
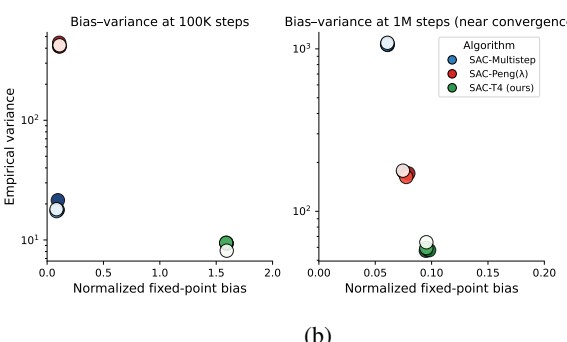

(a)                                                                     (b)

Figure 1: **Effect of truncation horizon on multi-step off-policy RL.** (a) Hopper performance with cap lengths $[3, 5, 10, 20]$ for `T4`, `Peng's` $Q(\lambda)$, and uncorrected $n$-step (darker→shorter caps), showing that longer horizons amplify off-policy errors in baseline methods. (b) Bias–variance patterns at 0.1M and 1M steps, where only `T4` maintains low variance through its adaptive truncation based on the estimated meeting time.

identify a key underlying cause: the lack of principled trajectory truncation, which yields cumulative errors through the product of *per-step* correction ratios and the *residual tail*. Through controlled experiments on `MuJoco Hopper` in Figure 1a, we empirically show that these cumulative errors scale rapidly with horizon length, leading to unreliable policy evaluation and degraded performance. In practice, multi-step methods rarely use the full episode length but instead define a maximum *cap length* as an upper bound on the truncation horizon, thereby introducing a hyperparameter that is often difficult to tune.

This challenge can be further understood through the theoretical framework of (Duan et al., 2024), which connects horizon selection to the mixing time of the underlying Markov Decision Process (MDP) and to model misspecification in value function approximation. However, estimating an appropriate horizon online is non-trivial, since the mixing time is difficult to measure on the fly.

To overcome this difficulty, we propose a *stochastic and adaptive truncation* mechanism within the Retrace framework (Munos et al., 2016), which we call `T4` *(Time To Truncate Trajectory)*. We estimate an upper bound on the mixing time via a coupling-based analysis of the Markov chains induced by the behavior and target policies (Johndrow & Mattingly, 2017a). This bound then guides our adaptive truncation strategy and enables `T4` to balance the trade-off between bias and variance. Theoretically, we prove that `T4` is non-conservative yet converges without imposing restrictions on behavior policy updates. Unlike prior multi-step methods that require careful cap length tuning, `T4` is robust to this hyperparameter and requires minimal tuning. Despite its simplicity, we show that `T4` consistently improves policy evaluation.

**Contributions.** Our main contributions are threefold. First, we demonstrate that naïve extension of the truncation horizon (cap length) amplifies cumulative errors in off-policy multi-step RL. Second, we connect horizon selection to mixing time and derive an approximate upper bound via the coupling argument to guide adaptive truncation, validating this both theoretically and empirically. Third, we propose `T4`, a stochastic and adaptive truncation method built upon the Retrace framework, and establish both its convergence guarantees and strong empirical performance.

## 2 PRELIMINARIES

We consider a MDP defined by the tuple $(\mathcal{S}, \mathcal{A}, \mathcal{P}, \mathcal{P}_0, \mathcal{R}, \gamma)$, where $\mathcal{S} \subset \mathbb{R}^d$ is a finite state space, $\mathcal{A}$ is a finite action space, $\mathcal{P} : \mathcal{S} \times \mathcal{A} \to \Delta(\mathcal{S})$ is the transition probability mapping each state-action pair to a distribution over next states, $\mathcal{P}_0 : \mathcal{S} \to [0, 1]$ is the initial state distribution, $\mathcal{R} : \mathcal{S} \times \mathcal{A} \to [-r_{\max}, r_{\max}]$ is a uniformly bounded reward function, and $\gamma \in [0, 1)$ is a discount factor for the infinite-horizon RL setting. Given a policy $\pi$, we define the Bellman operator as $\mathcal{T}^\pi Q := \mathcal{R} + \gamma \mathcal{P}^\pi Q$, where $\mathcal{P}^\pi$ denotes the transition operator induced by the environment dynamics $\mathcal{P}$ and the policy $\pi$. We use trajectories $(s_t, a_t, r_t)_{t \geq 0} \sim \beta$, where $\beta(\cdot \mid s)$ is *behavior policy*.

Since we focus on multi-step off-policy RL, we consider $K$-step off-policy evaluation using trajectories $(s_t, a_t, r_t)_{t \geq 0}$ generated by the behavior policy $\beta$. Specifically, we apply the $(k-1)$-fold composition of the Bellman operator for the behavior policy $\beta$, denoted by $\mathcal{T}^{\beta^{(k-1)}} : \mathbb{R}^{\mathcal{S} \times \mathcal{A}} \to \mathbb{R}^{\mathcal{S} \times \mathcal{A}}$, for $k = 1, \ldots, K$. We define *uncorrected $K$-step return operator* at iteration $n$ as

$$Q_{n+1} = \underbrace{r_t + \gamma r_{t+1} + \cdots + \gamma^{K-1} r_{t+K-1}}_{\text{from a behavior policy } \beta} + \gamma^K \mathcal{P}^\pi Q_n = \mathcal{T}^{\beta^{(K-1)}} \mathcal{T}^\pi Q_n. \tag{1}$$

**General Retrace.** One of the main challenges in multi-step off-policy RL is that policy evaluation can suffer from fixed-point bias (Munos et al., 2016) caused by the discrepancy between the target and behavior policies (Rowland et al., 2020). To correct this discrepancy, Munos et al. (Munos et al., 2016) proposed the *general Retrace* formulation, which addresses the fixed-point bias in off-policy evaluation by introducing a sequence of correction coefficients, referred to as *traces*. We formally define the *general Retrace operator* $\mathcal{R}$, which corrects the distributional discrepancy arising in off-policy evaluation:

$$\mathcal{R} Q_n = Q_n + \mathbb{E}_\beta \left[ \sum_{t=0}^{\infty} (\gamma \lambda)^t \left( \prod_{i=1}^{t} c(s_i, a_i) \right) (r_t + \gamma \mathbb{E}_{\pi_n}[Q_n(s_{t+1}, \cdot)] - Q_n(s_t, a_t)) \right], \tag{2}$$

where the sequence $\{c(s_i, a_i)\}$ is referred to as the *trace*, with the convention that $\prod_{i=1}^{0} c(s_i, a_i) = 1$ for $t = 0$. Here, $\pi_n$ denotes the target policy at the $n$-th iteration, and the formulation also incorporates a $\lambda$-extension (Bertsekas & Ioffe, 1996), which smoothly interpolates between $K$-step returns and the full Monte Carlo return. Multi-step off-policy RL algorithms can be expressed within the general Retrace by specifying the trace. Depending on the choice of $c_i$, these algorithms can be categorized into *conservative* and *non-conservative* methods. An algorithm is referred to as *conservative* if it satisfies $0 \leq c_i \leq \frac{\pi_n(a_i|s_i)}{\beta(a_i|s_i)}$ for all $i$. Conservative methods prevent overestimation through the trace constraint, thus their convergence are not affected by the update rule of the behavior policy $\beta_n$.

**Mixing time and Truncation Length.** While the standard retrace does not truncate the trajectories, in practice, the choice of a *truncation length* plays a critical role in learning performance (Hessel et al., 2018; Kozuno et al., 2021). In particular, longer truncation lengths can amplify the distributional discrepancy between the behavior and target policies, thereby degrading the accuracy of off-policy evaluation. We begin by defining the *stationary distribution* and *mixing time*. The key to our analysis is to connect truncation lengths with the mixing time of the MDP under $\mathcal{P}^\beta$.

The stationary distribution $\mu_\beta$ of the transition dynamics $\mathcal{P}^\beta$ is defined as the unique distribution to which the $t$-step state visitation distribution converges, i.e., $\mathcal{P}^{\beta^{(t)}}(s_1, s_2) \to \mu_\beta(s_2)$ as $t \to \infty$ for all $s_1, s_2 \in \mathcal{S}$. To analyze convergence to the stationary distribution, we introduce the notion of *coupling*. Given two distributions $\nu_1$ and $\nu_2$ over $\mathcal{S}$, a probability distribution $\omega$ over $\mathcal{S} \times \mathcal{S}$ is called a *coupling* of $\nu_1$ and $\nu_2$ if its marginals satisfy $\nu_1(x) = \sum_{y \in \mathcal{S}} \omega(x, y)$ and $\nu_2(y) = \sum_{x \in \mathcal{S}} \omega(x, y)$.

The *mixing time* $\tau_{\text{mix}}$ of $\mathcal{P}^\beta$ is defined as the smallest time $t$ at which the total variation distance between the $t$-step transition distribution and the stationary distribution becomes smaller than a threshold $\epsilon > 0$:

$$\tau_{\text{mix}} := \max_{s \in \mathcal{S}} \min \left\{ t : \left\| \mathcal{P}^{\beta^{(t)}}(s, \cdot) - \mu_\beta \right\|_{\text{TV}} \leq \epsilon \right\}. \tag{3}$$

Recent work by Duan et al. (Duan et al., 2024) established theoretical conditions for selecting the truncation length in infinite-horizon $\gamma$-discounted MDPs to improve the sample complexity of policy evaluation. Specifically, they derived a lower bound on the truncation length $K$ that controls the estimation error of an approximate $Q$-function. For uniformly bounded rewards, this bound takes the form

$$K = \min \left( \frac{1}{1 - \gamma}, \Omega(\tau_{\text{mix}}) \right), \tag{4}$$

where the notation $\Omega(\cdot)$ denotes an asymptotic lower bound, implying that $K$ must scale at least proportionally to the mixing time $\tau_{\text{mix}}$. The term $1/(1 - \gamma)$ corresponds to the standard discount-determined effective horizon.

This bound provides a principled guideline for choosing $K$, but its practical use is limited: estimating $\tau_{\mathrm{mix}}$ during learning is notoriously difficult because the full transition kernel of the behavior policy $\mathcal{P}^\beta$ is not observable online (Wolfer & Kontorovich, 2019).

**Paper Organization.**   In Section 3, we introduce our main contribution, the stochastic operator T4, and establish its convergence properties. T4 is designed not only as a stochastic extension of Retrace, but also as a mechanism to adaptively estimate the truncation horizon during learning. Section 4 then connects trajectory truncation with mixing-time upper bounds, showing how the disagreement probabilities encoded in T4 provide a principled way to approximate the mixing time of the behavior policy and thus determine an appropriate truncation length without requiring direct access to the mixing time itself.

## 3   TIME TO TRUNCATE TRAJECTORY (T4) OPERATOR

Our goal is to estimate the target value function $Q^\pi(s, a)$ from trajectories generated by an arbitrary behavior policy $\beta$. Beyond policy evaluation, we further show that T4 converges to the optimal value function $Q^*(s, a)$ under arbitrary behavior policies. To connect trajectory truncation with the general Retrace framework, we define a sequence of Bernoulli random variables $(A_i)$ corresponding to the trace coefficients in Equation (2), with associated probabilities $p = (p_1, p_2, \ldots)$. For each step $i$, let $S_i^\beta \sim \mathcal{P}_0(\mathcal{P}^\beta)^i$ and $S_i^\pi \sim \mathcal{P}_0(\mathcal{P}^\pi)^i$ denote the $i$-step state random variables generated respectively by the behavior policy $\beta$ and the target policy $\pi$, starting from the same initial distribution $\mathcal{P}_0$. Each $A_i$ then acts as an indicator of mismatch:

$$A_i = \mathbf{1}\{S_i^\beta \neq S_i^\pi\}, \qquad p_i := \Pr(A_i = 1) = \Pr(S_i^\beta \neq S_i^\pi) = \mathbb{E}[A_i]. \tag{5}$$

By replacing the deterministic trace coefficients $c_i$ in Equation (2) with the Bernoulli indicators $A_i$, we obtain the stochastic version of the Retrace operator, which we refer to as the T4 operator:

$$\mathcal{R}_{p,\lambda} Q \;=\; Q + \mathbb{E}_{\beta,p}\left[\sum_{t=0}^\infty \gamma^t \Big(\prod_{i=1}^t \lambda A_i\Big)\big(r_t + \gamma \mathbb{E}_\pi Q(s_{t+1}, \cdot) - Q(s_t, a_t)\big)\right]. \tag{6}$$

Once $A_i = 0$ for the first time, all subsequent terms vanish. Whereas previous multi-step RL approaches terminate the return at a fixed cap length—typically the episode length or a manually chosen horizon—our method stochastically adapts the truncation point.

We now aim to establish a lower bound on the truncation length $K$ in Equation (4) for off-policy RL. Since off-policy learning involves both a behavior policy $\beta$ and a target policy $\pi$, we upper bound the total variation $\max_{s \in \mathcal{S}} \|\mathcal{P}^{\beta^{(t)}}(s, \cdot) - \mu_\beta\|_{\mathrm{TV}}$ using the discrepancy between the transition kernels $\mathcal{P}^\beta$ and $\mathcal{P}^\pi$. In this setting, the mixing time is related to the total variation distance, which we analyze in Section 4. Here, we estimate this quantity via the sampled Bernoulli variables in Equation (5), where $\Pr(S_i^\beta \neq S_i^\pi)$ represents the one-step discrepancy between the behavior and target policies. This discrepancy is exactly the total variation distance between the induced state-transition distributions[2]. Hence, it can be expressed as

$$p_i := \Pr(A_i = 1) = 1 - \sum_{s' \in \mathcal{S}} \min\left\{\sum_a \beta(a \mid s_i)\mathcal{P}(s' \mid s_i, a), \; \sum_a \pi(a \mid s_i)\mathcal{P}(s' \mid s_i, a)\right\}. \tag{7}$$

Before we present the theoretical relation between truncation length and mixing time in Section 4, we first show that the T4 operator is a contraction mapping in the off-policy evaluation setting.

**Theorem 1** (Contraction of $\mathcal{R}_{p,\lambda}$). *Suppose $p_i \leq \xi$ almost surely for some $\xi \in [0, 1]$ and all $i \geq 1$. If $\gamma \in (0, 1)$ and $\lambda \in [0, 1]$ further satisfy*

$$\gamma < \frac{1}{1 + \xi}, \qquad \lambda \leq \min\left\{1, \frac{1 - \gamma(1 + \xi)}{2\gamma^2 \xi^2}\right\}, \tag{8}$$

---

[2]This follows from the maximal coupling lemma; see Appendix A for a formal proof and further discussion.

*then for any Q-function, the operator $\mathcal{R}_{p,\lambda}$ in Equation (6) has a unique fixed point $Q^\pi$ and satisfies*

$$\|\mathcal{R}_{p,\lambda}Q - Q^\pi\|_{\infty,p} \leq \eta(\gamma,\lambda,\xi)\|Q - Q^\pi\|_{\infty,p}, \tag{9}$$

*with contraction modulus*

$$\eta(\gamma,\lambda,\xi) = \frac{\gamma}{1-\gamma\xi} + \frac{2\lambda\gamma^2\xi^2}{1-\gamma\xi} < 1, \tag{10}$$

*where $\|\cdot\|_{\infty,p}$ denotes the supremum norm weighted by $p$.*

The proof is in Appendix E.

**Remark.** The assumption in Theorem 1 that $p_i \leq \xi$ is mild, since $p_i$ is a Bernoulli probability and thus always lies in $[0, 1]$. The bound merely introduces a uniform constant $\xi \leq 1$, with the trivial choice $\xi = 1$ always valid. Smaller values of $\xi$ yield a sharper contraction modulus in Equation (10).

We note that Retrace enforces $0 \leq c_i \leq \pi(a_i|s_i)/\beta(a_i|s_i)$, ensuring that each update is a sub-convex combination and thus strictly conservative. In contrast, T4 requires only the weaker condition $p_t \leq \xi$ while still guaranteeing contraction. This relaxation provides greater flexibility, enabling non-conservative updates without sacrificing convergence guarantees.

Theorem 1 shows that the T4 operator is a contraction mapping in the policy evaluation setting, converging to the fixed point $Q^\pi$. We next turn to the control setting, where the target policy is updated online. As in Retrace, no restrictive assumptions on the behavior policies are required; under arbitrary behavior policies, T4 converges to the optimal value function $Q^*$.

**Theorem 2** (Convergence in online control). *Let a sequence of Q-functions $(Q_n)$ be updated by the T4 operator, i.e.,*

$$Q_{n+1} = \mathcal{R}_{p,\lambda}Q_n.$$

*For arbitrary sequences of behavior policies $(\beta_n)$ and target policies $(\pi_n)$, we have $Q_n \to Q^*$ in the online control setting.*

The proof is in Appendix F. Together, Theorems 1 and 2 establish that T4 achieves reliable convergence both in policy evaluation and online control. We next analyze the relation between truncation length and mixing time, which underpins the construction of the Bernoulli probabilities $p_i$.

## 4 TRUNCATION LENGTH VIA MIXING-TIME UPPER BOUNDS

We now establish how the T4 operator provides a mechanism to approximate the mixing time of the behavior policy $\beta$ by relating trajectory truncation to discrepancies between transition kernels. To this end, we introduce formal quantities that characterize the discrepancy between the transition kernels of the behavior policy $\mathcal{P}^\beta$ and the target policy $\mathcal{P}^\pi$. Background on total variation distance and the coupling lemma, which underpin our analysis here, is summarized in Section A. Now, we formalize the notion of how far the two policy-induced kernels can differ at each state.

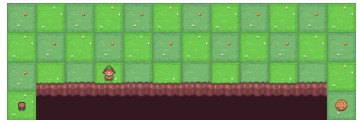

Figure 2: `CliffWalking`. A simple tabular environment with absorbing cliff dynamics.

**Definition 1** (Uniform $d$-bounded kernel). We say that the transition kernel $\mathcal{P}^\beta$ of a behavior policy $\beta$ is **uniformly $d$-bounded** if there exists $d \in (0, 1)$ such that for all states $s \in \mathcal{S}$ and any target policy $\pi$,

$$\left\|\mathcal{P}^\beta(s,\cdot) - \mathcal{P}^\pi(s,\cdot)\right\|_{\mathrm{TV}} \leq d.$$

This condition ensures that the transitions do not change drastically across policies, enabling the analysis of policy discrepancies. The notion of *perturbed Markov chains* is closely related to this setting, where transition kernels under different policies can be viewed as small perturbations of a given kernel. Such assumptions have been widely used in approximate Markov chain Monte Carlo (MCMC) (Mitrophanov, 2005; Solan & Vieille, 2003; Johndrow & Mattingly, 2017b).

**Assumption 1** (Cross-Doeblin Condition). There exists a constant $\rho \in (0, 1 - d)$ such that, for all states $s, s'$ and any policies $\beta, \pi$, $\left\|\mathcal{P}^\beta(s,\cdot) - \mathcal{P}^\pi(s',\cdot)\right\|_{\mathrm{TV}} \leq 1 - \rho$.

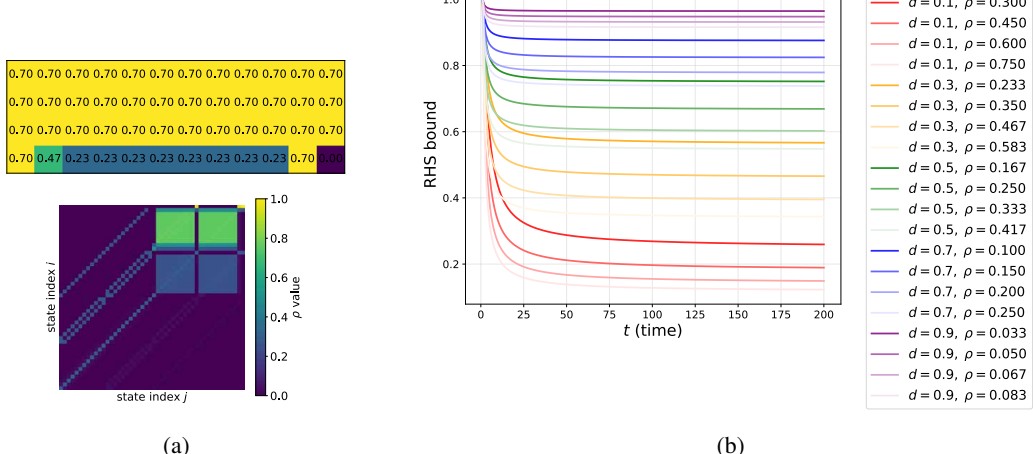

(a)                                                    (b)

Figure 3: (a) **Diagnostics on** `CliffWalking`: state-wise total variation distance $d(s)$ and pairwise overlap matrix $\rho(s, s')$ between the transitions induced by an optimal policy and a uniformly random policy. Large $d$ and near-zero $\rho$ across most states highlight the structural off-policy gap. (b) **RHS bounds from Lemma 2** for multiple $(d, \rho)$ pairs: the bounds consistently decay across all settings, implying that a meeting time emerges even under severe off-policy mismatch.

The cross-Doeblin condition plays a central role in assessing the approximation quality of MCMC algorithms (Mattingly et al., 2015; Johndrow & Mattingly, 2017a). In our context, it serves as a regularity assumption ensuring that the transition distributions under any pair of states and policies are sufficiently close. This allows us to model the target transition kernel $\mathcal{P}^\pi$ as a perturbation of the behavior kernel $\mathcal{P}^\beta$, thereby facilitating the estimation of the mixing time of the behavior policy.

Although Definition 1 and Assumption 1 may appear strong, they represent the weakest meaningful conditions that allow us to quantify the kernel-level discrepancies required for estimating the mixing time of the underlying Markov chains. Our diagnostic study on the tabular `CliffWalking` environment in Figure 2 highlights this point: even slight deviations between $\pi$ and $\beta$ lead to nearly deterministic branching and absorbing transitions (falling off the cliff), pushing state-wise TV distances close to 1 and collapsing cross-state overlaps. This indicates that $d$ and $\rho$ are largely determined by structural properties of the MDP rather than by policy proximity.

To make this explicit, Figure 3a reports the state-wise TV distances $d(s)$ and the pairwise overlap matrix $\rho(s, s')$ between the transition kernels of an optimal policy and a uniformly random policy. Both quantities exhibit extreme mismatch—large $d$ and near-zero $\rho$ across most states—revealing a substantial structural off-policy gap even in this simple tabular setting. Nevertheless, we show below that our subsequent analysis remains valid despite these harsh structural properties. We now turn to present the key lemmas and theorems that characterize how these quantities govern disagreement probabilities, meeting times, and the resulting effective truncation length.

**Lemma 1.** *For a given behavior policy $\beta$ and transition kernel $\mathcal{P}^\beta$ which is uniformly ergodic with $\alpha$, let $\mu_\beta$ denote the stationary distribution of $\mathcal{P}^\beta$. Then, for any policy $\pi$ and initial state $s$, we have*

$$\left\|\mathcal{P}^{\beta(t)}(s, \cdot) - \mu_\beta\right\|_{\text{TV}} \le \left\|\mathcal{P}^{\beta(t)}(s, \cdot) - \mathcal{P}^{\pi(t)}(s, \cdot)\right\|_{\text{TV}} + 1 - \alpha + \frac{d}{\alpha}. \tag{11}$$

The proof is in Appendix G. Lemma 1 offers insight into how the convergence of $\mathcal{P}^\beta$ is related to the discrepancy between $\mathcal{P}^{\beta(t)}$ and $\mathcal{P}^{\pi(t)}$. By the coupling lemma (see Appendix A), the total variation between two transition kernels is at most the probability that the coupled variables disagree; in our notation, $\left\|\mathcal{P}^{\beta(t)}(s, \cdot) - \mathcal{P}^{\pi(t)}(s, \cdot)\right\|_{\text{TV}} \le \mathrm{P}(S_t^\beta \ne S_t^\pi)$.

**Lemma 2.** *Let $S_k^\beta \sim \mathcal{P}_0(\mathcal{P}^\beta)^k$ and $S_k^\pi \sim \mathcal{P}_0(\mathcal{P}^\pi)^k$ with the initial state distribution $\mathcal{P}_0$ as the random variables corresponding to the $k$-step state distributions. Let $A_k$ be the Bernoulli indicator*

*defined in Equation* (5), *i.e.,* $A_k = \mathbf{1}\{S_k^\beta \neq S_k^\pi\}$ *with* $\Pr(A_k = 1) = \Pr(S_k^\beta \neq S_k^\pi)$. *Then,*

$$\frac{1}{t}\sum_{k=1}^{t}\mathbb{E}[A_k] \;\leq\; \frac{d}{\rho+d} + \frac{1-(1-\rho-d)^t}{t(\rho+d)}\left(\mathbb{E}[A_1] - \frac{d}{\rho+d}\right). \tag{12}$$

The proof is in Appendix H. Lemma 2 establishes that the time-average probability of disagreement between the two coupled processes decays over time. In Figure 1b, we plot the RHS bounds from Lemma 2 for several $(d, \rho)$ pairs. Across all configurations, the bounds decay steadily, indicating that the coupled processes still admit a finite meeting time even under severe off-policy mismatch.

Equivalently, this suggests that the processes eventually coalesce with high probability, and the relevant notion of convergence is captured by the *first meeting time* between them. This motivates introducing the random variable $T_{\beta,\pi}$, which directly quantifies the expected horizon until the two trajectories align. We now show how this notion provides a principled way to determine the effective truncation length.

**Theorem 3.** *Let the random variable* $T_{\beta,\pi}$ *denote the first meeting time of two processes, defined as*

$$T_{\beta,\pi} := \min\left\{t \geq 1 : S_t^\beta = S_t^\pi \mid S_0 \sim \mathcal{P}_0\right\}. \tag{13}$$

*The random variable* $T_{\beta,\pi}$ *can then be used to refine the truncation length condition in Equation* (4), *leading to the following formulation:*

$$K = \min\left(\frac{1}{1-\gamma}, \; \mathbb{E}[T_{\beta,\pi}]\right). \tag{14}$$

*That is, the effective truncation length is determined by either the discount horizon* $1/(1-\gamma)$ *or the expected meeting time* $\mathbb{E}[T_{\beta,\pi}]$, *whichever is smaller.*

**Remark.** By coupling arguments, the expected meeting time $\mathbb{E}[T_{\beta,\pi}]$ provides a lower bound on the mixing scale, i.e., $\mathbb{E}[T_{\beta,\pi}] = \Omega(\tau_{\mathrm{mix}})$. Thus, the truncation length in Equation (14) is always at least on the order of the intrinsic mixing time of the underlying Markov chain.

The proof and the formal connection between $\mathbb{E}[A_t]$ and $\mathbb{E}[T_{\beta,\pi}]$ are in Section I. We first note that the expectation $\mathbb{E}[T_{\beta,\pi}]$—the first meeting time between the two processes—can be estimated by sampling the time until the first match from $t = 0$. Let $t'$ denote the first time step such that $S_{t'}^\beta = S_{t'}^\pi$, which implies $A_{t'} = 0$. Since this is the first agreement point, we have $\prod_{i=1}^{t'} A_i = 0$. This construction leads to a natural truncation mechanism in the $\mathtt{T4}$ operator: for all $t \geq t'$, the temporal-difference (TD) error is set to zero, effectively stopping the credit assignment beyond the first matching point. Specifically, we have

$$\left(\prod_{i=1}^{t}\lambda A_i\right)(r_t + \gamma\mathbb{E}_\pi Q(s_{t+1}, \cdot) - Q(s_t, a_t)) = 0 \quad \text{for } t \geq t' \tag{15}$$

This truncation reflects the assumption that once the trajectories align, their future evolution can be treated as equivalent, thereby eliminating the need for further correction beyond the meeting time.

## 4.1 PRACTICAL IMPLEMENTATION

Building on the theoretical results from Sections 3 and 4, we now present a practical instantiation of the $\mathtt{T4}$ operator that computes the truncation length. The goal is to mitigate distributional discrepancy between the target and behavior policies and thereby reduce off-policy evaluation error.

**Approximating disagreement probabilities.** In theory, the Bernoulli variables $A_i$ are defined through $p_i = \Pr(S_i^\beta \neq S_i^\pi)$ in Equation (5), which requires access to the transition kernel $\mathcal{P}$. Since this is unavailable in the *model-free RL*, we approximate $p_i$ by measuring the overlap between the two policies on the sampled action $a_i$:

$$\hat{p}_i = 1 - \min\{\beta(a_i \mid s_i), \pi(a_i \mid s_i)\}. \tag{16}$$

This proxy interprets the shared support of $\beta$ and $\pi$ at $(s_i, a_i)$ as the agreement probability, with its complement serving as a model-free estimate of disagreement. A key structural fact is that the environment transition kernel is policy-independent. Thus, $P_\beta(\cdot \mid s)$ and $P_\pi(\cdot \mid s)$ are obtained by pushing $\beta(\cdot \mid s)$ and $\pi(\cdot \mid s)$ through the same kernel, which implies a data-processing inequality:

$$p_i = \mathrm{TV}(P_\beta(\cdot \mid s_i), P_\pi(\cdot \mid s_i)) \leq \mathrm{TV}(\beta(\cdot \mid s_i), \pi(\cdot \mid s_i)).$$

Although $\hat{p}_i$ is a noisy approximation, it preserves the correct monotonic dependence on policy mismatch and provides a practical surrogate for the theoretical $p_i$ used in our meeting-time analysis. A detailed justification and formal derivation are provided in Appendix J.

**Sampling the meeting time.** Using these estimates, we form stochastic traces $\hat{A} = (\hat{A}_1, \hat{A}_2, \ldots)$ with $\hat{A}_i \sim \mathrm{Bernoulli}(\hat{p}_i)$. The estimated meeting time $\hat{T}_{\beta,\pi}$ is taken as the first index $t$ for which $\hat{A}_t = 0$, and the truncation length is then defined as

$$\hat{K} = \min\big\{\lceil (1-\gamma)^{-1}\rceil, \hat{T}_{\beta,\pi}, \big\},$$

We also enforce $\hat{K} \geq 1$ to avoid trivial truncations.

**Integration with standard algorithms.** The pseudocode in Algorithm 1 shows how T4 modifies a generic actor-critic update such as SAC (Haarnoja et al., 2018) or TD3 (Fujimoto et al., 2018).

The only difference lies in lines 8–9, where each sampled history trajectory $h_i$ is *explicitly truncated* at length $K$. The explicit stochastic truncation mechanism in T4 has two key benefits. First, it avoids variance amplification from long products of importance weights, since trajectories are truncated immediately after the first meeting point. Second, it reduces sensitivity to manually chosen cap lengths: the effective horizon is adaptively determined by either the discount horizon $(1-\gamma)^{-1}$ or the estimated meeting time $\hat{T}_{\beta,\pi}$, whichever is smaller.

---

**Algorithm 1** Time to Truncate Trajectory (T4).

---

1: Initialize Q-function $Q_\theta$, target policy $\pi_\phi$, behavior policy $\beta_\phi$
2: $\mathcal{B} \leftarrow$ empty replay memory.
3: **for** each episode **do**
4:     **for** each step **do**
5:         Observe $s$ and take $a \sim \beta_\phi$
6:         Get next state $s' \sim \mathcal{P}(s, a)$ and reward $r$
7:         Store $\{(s, a, r, s')\}$ in $\mathcal{B}$
8:         Sample history minibatch $\{h_i\}_{i=1}^B \sim \mathcal{B}$
9:         Truncate $h_i$ with $\hat{K} = \min\{(1-\gamma)^{-1}, \hat{T}_{\beta,\pi}\}$.
10:         Update $\theta$ and $\phi$
11:     **end for**
12: **end for**

---

## 5 EXPERIMENTS

We evaluate T4 under both SAC and TD3 backbones, and compare against four baseline methods: the original one-step algorithm, an uncorrected $n$-step variant, Retrace (Munos et al., 2016), and Peng's $Q(\lambda)$ (Kozuno et al., 2021). All methods use identical network architectures and hyperparameters as their one-step baselines to ensure fair comparison. Detailed update rules and full hyperparameter settings are provided in Appendix C.

Figure 4 compares SAC,TD3-T4 with four multi-step baselines across five MuJoCo tasks. SAC-T4 consistently achieves strong performance and converges faster than the baselines. SAC-Retrace which is a conservative method performs comparably to T4 only on humanoid-v2 but lags behind elsewhere. Non-conservative methods (Peng's $Q(\lambda)$ and $n$-step) show mixed results and often underperform even the one-step SAC baseline. Additional TD3-based results are reported in Figure 7 of Appendix. We also report the adaptive truncation lengths computed by T4, shown in the lower-right panel of Figure 4. These results indicate that fixed $n$-step baselines can suffer when the effective truncation horizon is shorter than the chosen cap length $n$, while T4 remains stable. An ablation on the choice of truncation length in Figure 5 (top) further confirms that T4 is robust to this hyperparameter.

**Evaluation protocol.** For fair comparison, we primarily follow standard practice in off-policy RL benchmarks. In addition, we also report results under the more robust evaluation protocol of

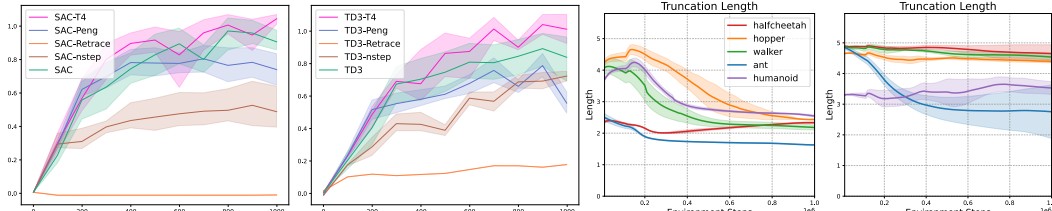

Figure 4: **Performance with stochastic truncation.** We report IQM-normalized scores across five MuJoCo tasks for SAC- and TD3-based methods, showing that T4 consistently outperforms multi-step baselines and converges faster. The right panels visualize the adaptive truncation lengths estimated by T4 for SAC (third) and TD3 (fourth), illustrating how the effective horizon contracts as the target policy aligns with the behavior policy. See Section D.1 for more information.

(Agarwal et al., 2021), which computes interquartile mean (IQM) normalized scores. As discussed in Figure 4, this protocol further highlights the efficiency of T4, showing that it surpasses expert-level performance in MuJoCo tasks significantly faster than competing multi-step methods.

## 6 DISCUSSION

**Truncation length should be adaptive.** Our results highlight that the key difficulty in multi-step off-policy RL lies in choosing an appropriate truncation horizon as illustrated in Figure 1a. When trajectories are sampled from a sequence of changing behavior policies, the effective horizon depends not only on the discount factor but also on the mismatch between the behavior and target policies. Thus, treating the truncation length $K$ as a fixed *cap length*, as in conventional $n$-step methods, is inherently problematic. This observation is consistent with prior empirical findings in both model-free (Rainbow) and model-based (MBPO) papers, where adaptive horizons improved stability.

**When Long Horizons Are Needed (large $K$).** A large effective horizon arises when the behavior policy mixes slowly or explores regions of the state space that the target policy has not yet adapted to. In this case, the expected meeting time between trajectories is long, and algorithms that fix $n$ too small (e.g., $n = 1$) lose useful long-horizon information. This explains why one-step SAC lags behind SAC-T4 in most environments: T4 adapts to maintain longer horizons (Figure 4). It can also be interpreted that, in such long-horizon regimes, the conservative trace coefficients of Retrace cut the updates too aggressively, discarding useful information and thereby degrading performance.

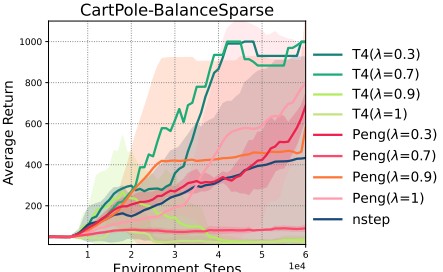

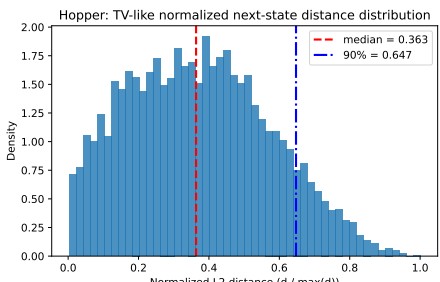

Figure 5: (Top) A sparse-reward control tasks from the DeepMind Control Suite. (Bottom) Normalized next-state discrepancy between transitions on `Hopper`.

**When Short Horizons Suffice (small $K$).** Conversely, as policy improvement aligns the target policy more closely with the behavior distribution, the trajectories meet earlier and the effective horizon shrinks. In this regime, non-conservative methods like Peng's $Q(\lambda)$ or uncorrected $n$-step continue to propagate credit too far, leading to unstable updates. Our ablation in Figure 1a confirms that T4 remains robust even when the effective truncation length decreases during training.

**Efficiency in model-based and sparse-reward settings.** Beyond dense-reward benchmarks, T4 also demonstrates strong efficiency in both model-based comparisons and sparse-reward tasks. As shown in Figure 9, T4 rapidly matches the sample efficiency of SAC-based MBPO while remaining

entirely model-free. In addition, Figure 5-(top) highlights that T4 achieves near-optimal performance significantly faster than Peng's method and $n$-step baselines in sparse-reward control, a regime where prior success has mostly relied on model-based or skill-specific techniques.

**Bias–variance trade-off and empirical support for the TV-based analysis.** Our cap-length ablations in Figure 1 reveal a clear bias–variance trade-off consistent with the coupling-based view. As the truncation horizon increases, multi-step baselines such as uncorrected $n$-step and Peng's $Q(\lambda)$ accumulate off-policy discrepancies multiplicatively, producing high-variance and biased updates. This issue appears most clearly in Hopper, where performance degrades as the cap length increases from 3 to 20, reflecting the mismatch between fixed caps and the evolving behavior–target divergence. In contrast, T4 remains stable across all truncation lengths: stochastic truncation at the estimated meeting time removes long-tail variance while preserving essential multi-step information. The adaptive horizons chosen by T4 match the regime predicted by our coupling analysis—shorter when $\beta$ and $\pi$ differ early in training, and longer as they align—mirroring the decay of disagreement probabilities in Lemma 2 and providing empirical support for the mixing-time interpretation. Figure 5-(bottom) provides an empirical sanity check of kernel similarity in continuous spaces. Using a 3D PCA embedding of next-state transitions and normalized L2 distances as a proxy for kernel divergence, we observe that most $(P_\beta, P_\pi)$ transitions lie well below half of the maximum discrepancy (median $\approx 0.36$, 90th percentile $\approx 0.64$), even when $\beta$ is uniformly random. It means that the uniform $d$-boundedness and cross-Doeblin overlap are reasonably satisfied in MuJoCo dynamics.

## 7 RELATED WORK

**Return-based off-policy and multi-step methods.** Our work builds on return-based off-policy algorithms (Mahmood & Sutton, 2015; Munos et al., 2016; Harutyunyan et al., 2016; Precup, 2000; Daley & Amato, 2019) and analyses of stochastic TD learning under Markovian sampling (Bhandari et al., 2018; Mou et al., 2020). Prior multi-step approaches mitigate off-policy mismatch through (i) weight correction (e.g., Retrace, Tree-Backup, V-trace) (Munos et al., 2016; Precup, 2000; Rowland et al., 2020), (ii) conservative updates (Kozuno et al., 2021), (iii) eligibility-trace formulations (Singh & Sutton, 1996; van Hasselt et al., 2021; Daley et al., 2023; Gupta et al., 2024), and (iv) model-based imagination (Hafner et al., 2020; Janner et al., 2019). These methods differ in how they trade off bias and variance when propagating multi-step credit. Large-scale RL systems such as R2D2 (Kapturowski et al., 2018) and IMPALA (Espeholt et al., 2018) highlight the practical importance of stabilizing long multi-step returns (e.g., via V-trace) rather than adaptively adjusting horizons.

**Why multi-step evaluation is hard.** Even with small correction weights, long-tail contributions from later trajectory segments introduce error under Markovian sampling, often yielding oscillatory $Q$-functions (Kozuno et al., 2021) and slow error decay (Berthier et al., 2022). Fixed caps alleviate this but can be misaligned with the environment's mixing scale, causing under-utilized long-horizon signal or excessive variance. Beyond trace reweighting, resampling-based approaches include importance resampling (Schlegel et al., 2019), stationary-distribution corrections (Yuan et al., 2021; Yang et al., 2020), and covariate-shift correction (Gelada & Bellemare, 2019; Hallak & Mannor, 2017). T4 is complementary: instead of estimating precise ratios, it stochastically truncated returns based on estimated disagreement between behavior and target rollouts, reducing sensitivity to fixed caps while remaining compatible with standard actor–critic methods.

## 8 CONCLUSION

We presented T4, a stochastic variant of Retrace that adaptively truncates trajectories at the estimated meeting time. This mechanism mitigates off-policy discrepancies while preserving useful long-horizon credit, consistently improving over one-step and multi-step baselines across diverse RL benchmarks. Our analysis relies on a $d$-bounded kernel condition, which serves as a simplified form of uniform ergodicity. Although we do not explicitly verify this assumption in our benchmark environments, the empirical results suggest that T4 remains effective even without strict mixing guarantees. Future work includes extending T4 to model-based settings for tighter horizon control and developing practical diagnostics to adapt truncation length online.

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

APPENDIX

## A    BACKGROUND ON ERGODICITY AND MAXIMAL COUPLING

We now state the *coupling lemma*, which provides a tool for bounding the total variation distance.

For any two probability distributions $\nu_1$ and $\nu_2$ over $\mathcal{S}$, we define the total variation (TV) distance $\|\cdot\|_{\text{TV}}$ as

$$\|\nu_1 - \nu_2\|_{\text{TV}} := \frac{1}{2} \sum_{s \in \mathcal{S}} |\nu_1(s) - \nu_2(s)| = \max_{A \subset \mathcal{S}} |\nu_1(A) - \nu_2(A)|,$$

where the norm $\|\cdot\|$ corresponds to the $L_1$ metric. By definition, the TV distance takes values in the interval $[0, 1]$.

**Lemma 3** (Coupling Lemma). *Let $\nu_1$ and $\nu_2$ be two probability distributions over a finite space $\mathcal{S}$. Then there exists a coupling $(X, Y)$ of $\nu_1$ and $\nu_2$ such that*

$$\mathrm{P}(X \neq Y) \geq \|\nu_1 - \nu_2\|_{TV}.$$

*A coupling that achieves this equality is called a maximal coupling and can be written as*

$$\mathrm{P}(X \neq Y) = \|\nu_1 - \nu_2\|_{TV} = 1 - \sum_{s \in \mathcal{S}} \min(\nu_1(s), \nu_2(s)). \tag{17}$$

Maximal coupling minimizes the probability of disagreement $\mathrm{P}(X \neq Y)$ among all possible couplings of $\nu_1$ and $\nu_2$. Under this condition, we say the MDP is *uniformly ergodic* with $\alpha$ if there exists a constant $\alpha \in (0, 1)$ and $C > 0$ such that

$$\max_{s \in \mathcal{S}} \left\| \mathcal{P}^{\beta(t)}(s, \cdot) - \mu_\beta \right\|_{\text{TV}} \leq C(1 - \alpha)^t$$

for all $t \in \mathbb{N}$.

## B    EXTENDED RELATED WORK

A parallel line of work studies policy learning under uniform/geometric mixing or access to the stationary distribution (Meyn & Tweedie, 2012; Hao et al., 2020; Abbasi-Yadkori et al., 2019; Neu & Olkhovskaya, 2021), and leverages mixing-time-aware analyses in MDPs (Suttle et al., 2023; Wei et al., 2021). In contrast, our approach is model-free and does not assume direct access to stationary distributions or exact mixing times. Instead, T4 adapts the truncation horizon via a stochastic meeting-time proxy derived from policy overlap, aligning the effective multi-step depth with the evolving off-policy mismatch during training.

## C    IMPLEMENTATION DETAILS

In this section, we describe the full implementation details of T4. Following the standard practice in off-policy RL, we use the PyTorch version of the implementations in OpenAI SpinningUp (Achiam, 2018).

**Experimental Setup**    We compare T4 with four baseline methods, a conventional one-step method, uncorrected multi-step method, Retrace (Munos et al., 2016) and Peng's $Q(\lambda)$ (Kozuno et al., 2021). Given a randomly sampled trajectory $(s_0, a_0, r_0, s_1, a_1, r_1, s_2, \cdots)$, where $Q_{\theta^-}$ denotes the target $Q$-function, and $\tilde{a}_\phi(s_i)$ is a sample from $\pi_\phi(\cdot|s_i)$. The detailed targets for the $Q$-function of all algorithms are described in Table 1 in Section C. We note that all algorithms we used are based on actor-critic method and update the policy network only with the starting target at $(s_0, a_0)$. For example, SAC based methods update the parameter of policy networks by gradient ascent $\arg\max_\pi Q_{\text{target}}(s_0, \tilde{a}_\phi(s_0)) + \alpha \log \pi_\phi(\tilde{a}_\phi|s_0)$.

**Training and evaluation.**    For all algorithms, we use $[256, 256]$-sized multi-layer perceptrons (MLPs) for all neural networks. We train with 1M environment steps for openAI Mujoco and evaluate the agent every 1000 steps by using deterministic policy in 10 episodes.

**Implementations of multi-step operators.** We provide pseudocode for multi-step off-policy actor-critic deep RL algorithms

The multi-step target value can be computed recursively for a given trajectory $(s_0, a_0, r_0, s_1, a_1, r_1, \cdots)$. Let $Q_{\theta_1}, Q_{\theta_2}$ be two Q-function critic and $\hat{Q}_i$ be the target value estimate at environment step $i$. We can write

$$\hat{Q}_i = r_i + \gamma \min(\max_a Q_{\theta_1}(s_i, a), \max_a Q_{\theta_2}(s_i, a))$$

$$+ \gamma\lambda \left( \hat{Q}_{i+1} - \min(\max_a Q_{\theta_1}(s_i, a), \max_a Q_{\theta_2}(s_i, a)) \right).$$

For continuous action space, we approximate $\max_a Q_\theta(s_i, a)$ as $Q_\theta(s_i, \pi_\phi(s))$. Practically, we use a finite-length trajectory $(s_0, a_0, r_0, s_1, a_1, r_1, \cdots, s_c)$ where $c$ is the cap length of the trajectory.

Table 1: The details of the multi-step targets for baselines and our method for SAC. We note that T4 samples each $A_1, A_2, \cdots, A_{k-1}$ from the corresponding probabilities $\hat{p}_1, \hat{p}_2, \cdots, \hat{p}_{K-1}$.

| Algorithm | Update pseudo-code |
|---|---|
| One-step RL | $r_0 + \gamma(Q_\theta(s_1, \tilde{a}_\phi(s_1)) - \alpha \log \pi_\phi(\tilde{a}_\phi(s_1)|s_1))$ |
| Uncorrected $K$ | $\sum_{i=0}^{K-1} \gamma^i r_i + \gamma^K(Q_\theta(s_1, \tilde{a}_\phi(s_1)) - \alpha \log \pi_\phi(\tilde{a}_\phi(s_1)|s_1))$ |
| Retrace | $\sum_{i=0}^{K-1} \gamma^i (\prod_{j=1}^i c_j)(r_i + \gamma(Q_\theta(s_{i+1}, \tilde{a}_\phi(s_{i+1})) - \alpha \log \pi_\phi(\tilde{a}_\phi(s_{i+1})|s_{i+1}) - c_{i+1}Q_\theta(s_{i+1}, a_{i+1}))$ |
| Peng's $Q(\lambda)$ | $\sum_{i=0}^{K-1} (\gamma\lambda)^i (r_i + \gamma(1-\lambda)(Q_\theta(s_{i+1}, \tilde{a}_\phi(s_{i+1})) - \alpha \log \pi_\phi(\tilde{a}_\phi(s_{i+1})|s_{i+1})).$ |
| T4 | $\sum_{i=0}^{K-1} \gamma^i (\prod_{j=1}^i A_j)(r_i + \gamma(Q_\theta(s_{i+1}, \tilde{a}_\phi(s_{i+1})) - \alpha \log \pi_\phi(\tilde{a}_\phi(s_{i+1})|s_{i+1}) - A_{i+1}Q_\theta(s_{i+1}, a_{i+1}))$ |

**Methods and Hyperparameters.** We use two one-step RL algorithms, SAC and TD3 for the multi-step extension.

1. **Twin-Delayed Deep Deterministic Policy Gradient (TD3).** TD3 (Fujimoto et al., 2018) adopts the same training pipeline and neural network architecture as DDPG, but introduces several improvements to address overestimation bias in Q-learning. Specifically, TD3 uses two critic networks, denoted as $Q_{\theta_1}(s, a)$ and $Q_{\theta_2}(s, a)$, with independent parameter sets $\theta_1$ and $\theta_2$. This twin-critic design follows the principle of double Q-learning (van Hasselt, 2010), which mitigates the positive bias introduced by max operators in standard Q-learning updates.

2. **Soft Actor-Critic (SAC).** SAC (Haarnoja et al., 2018) also adopts the same training pipeline and architecture as DDPG and TD3, but introduces a fundamentally different objective based on maximum entropy reinforcement learning. The core idea of SAC is to augment the reward function with an entropy term that encourages exploration by discouraging the policy from collapsing to a deterministic distribution. Similar to TD3, SAC maintains two critic networks to reduce the overestimation bias present in standard actor-critic methods.

Basically, we adopt all default hyper-parameters from the code base in OpenAI SpinningUp. The cap length denotes the upper limit of the sub-trajectory length for the baseline algorithms, unocrrected $n$-step, Retrace, and PQL. We report the detailed values in the below.

**Experimental Details.** We implement `T4` and other baselines in PyTorch on top of the standard evaluation protocol of off-policy RL ealgorithms in Google Dopamine (Castro et al., 2018) We provide our full implementation and commands to reproduce our main results of `T4` at (https://anonymous.4open.science/r/t4-BD20).

Table 2: TD3 Hyperparameters

| Hyperparameter | Value |
|---|---|
| Actor learning rate | $1 \times 10^{-3}$ |
| Critic learning rate | $1 \times 10^{-3}$ |
| Batch size | 100 |
| Replay buffer size | $1 \times 10^6$ |
| Discount factor $\gamma$ | 0.99 |
| Polyak averaging coefficient ($\tau$) | 0.995 |
| Target policy noise (stddev) | 0.2 |
| Target noise clip | 0.5 |
| Policy update delay (frequency) | 2 steps |
| Exploration noise (initial stddev) | 0.1 |
| Action range | [-1, 1] |
| Start steps (before training begins) | 10000 |
| Max episode length | 1000 |
| cap length | 5 |
| lambda ($\lambda$) | 0.7 |

Table 3: SAC Hyperparameters

| Hyperparameter | Value |
|---|---|
| Actor learning rate | $1 \times 10^{-3}$ |
| Critic learning rate | $1 \times 10^{-3}$ |
| Entropy coefficient (initial $\alpha$) | 0.2 (fixed) |
| Batch size | 100 |
| Replay buffer size | $1 \times 10^6$ |
| Discount factor $\gamma$ | 0.99 |
| Polyak averaging coefficient ($\tau$) | 0.995 |
| Target update interval | Every 1 step |
| Automatic entropy tuning | Enabled |
| Start steps (before policy used) | 10000 |
| Action range | [-1, 1] |
| Max episode length | 1000 |
| cap length | 5 |
| lambda ($\lambda$) | 0.7 |

## D   ADDITIONAL EXPERIMENTAL RESULTS

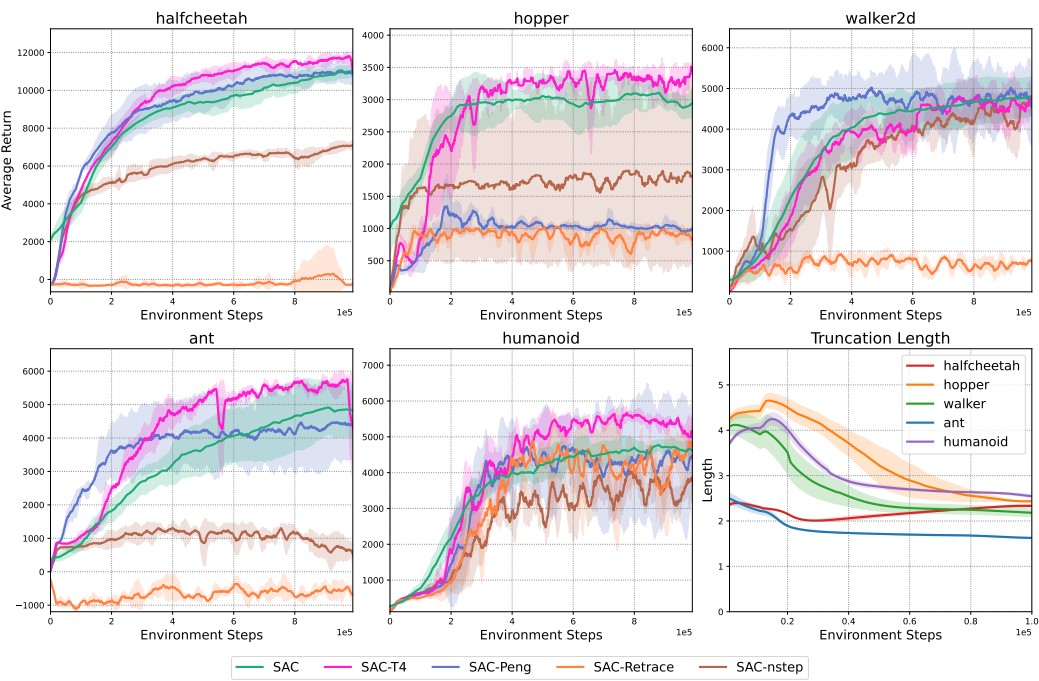

Figure 6: **Performance with stochastic truncation.** Across five MuJoCo benchmarks, our method (T4) consistently outperforms multi-step baselines and achieves faster convergence. For baseline comparisons, we follow the convention of (Kozuno et al., 2021) and fix the cap length to $n = 5$ for all multi-step methods.

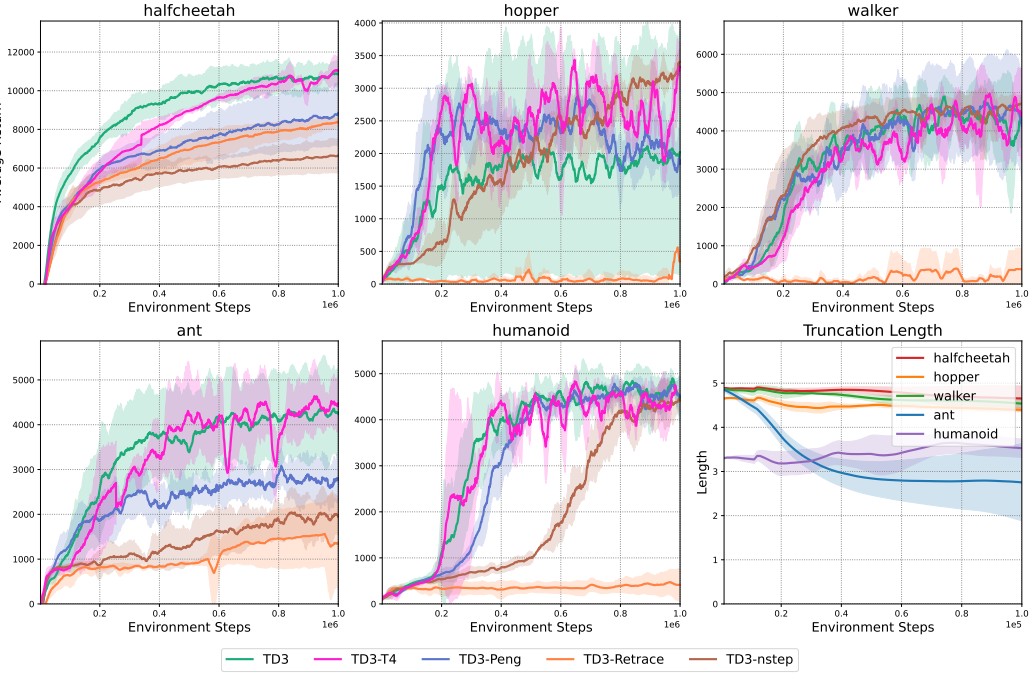

Figure 7: **Evaluation of Twin-Delayed Deep Deterministic Policy Gradients (TD3) variants over openAI mujoco environments.**

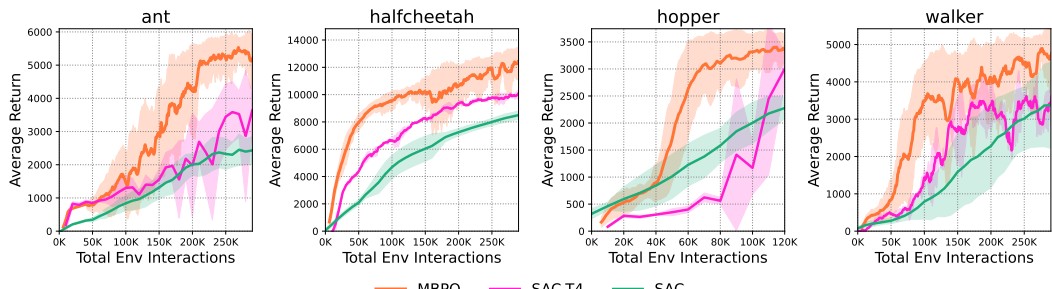

Figure 8: **Comparison of SAC, SAC-T4, and MBPO across four MuJoCo benchmarks.** Following our analysis of horizon sensitivity, we use different multi-step configurations per environment: *halfcheetah* and *walker* use a standard multi-step setting (cap length $= 5$, $\lambda = 0.7$), whereas *ant* and *hopper* follow the longer-horizon model-based configuration inspired by MBPO (cap length $= 25$, $\lambda = 0.1$). Across all tasks, T4 consistently accelerates learning and improves sample efficiency by adaptively adjusting its effective truncation length to the behavior–target policy mismatch, often matching or approaching MBPO despite being entirely model-free.

Table 4: **Reference scores for min–max normalization.** Random and expert performance values are taken from D4RL (Fu et al., 2021) and Minari (Farama Foundation, 2022). These values are used for computing normalized IQM scores in MuJoCo environments.

| Environment | Random Score | Expert Score |
|---|---|---|
| Hopper | $-20.27$ | 3234.3 |
| HalfCheetah | $-280.18$ | 12135.0 |
| Walker2d | 1.63 | 4592.3 |
| Ant | $-325.6$ | 3879.7 |
| Humanoid | 78.85 | 9024.95 |

### D.1  EVALUATION RESULTS BY THE PROTOCOL OF IQM

We conducted experiments following the evaluation protocol proposed in (Agarwal et al., 2021) to further examine the online RL performance of T4. This protocol emphasizes robust evaluation through interquartile mean (IQM) scores and normalized performance, and applying it highlights the superiority of our method in five different MuJoCo environments.

For these experiments, we computed min–max normalized scores across 10 runs (seeds). While prior work (Agarwal et al., 2021) typically reports 8 seeds, we adopted 10 runs to ensure more reliable estimates. The steps reported in the table correspond to 1,000K environment interactions. Metric computation was conducted using the official codebase of (Agarwal et al., 2021). For min–max normalization, expert and random scores were taken from the Minari extension (Farama Foundation, 2022) of D4RL (Fu et al., 2021); the Humanoid benchmark follows Minari scores, while the remaining tasks use D4RL values.

Across both SAC- and TD3-based experiments, T4 exhibits faster learning curves than existing multi-step methods and baseline algorithms. Notably, as shown in the table, only T4 surpasses the expert score within 1,000K steps, whereas prior methods typically require up to 3,000K steps to achieve comparable performance. This demonstrates the strong sample efficiency and effectiveness of our stochastic truncation approach.

## D.2 ADDITIONAL EARLY-TRAINING DIAGNOSTICS ON ATARI

To assess the stability of the proposed multi-step operator in the low-data regime, we report IQM results on `Pong` and `Breakout` at 500K agent steps (update horizon = 5, $\lambda = 1$). C51-T4 exhibits both higher normalized scores and reduced variance compared to C51, Rainbow, and DQN, indicating that stochastic truncation improves the robustness of multi-step distributional learning during early training.

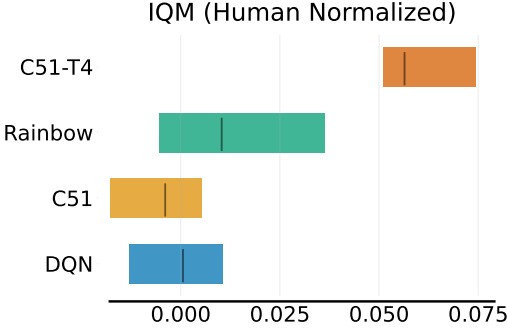

Figure 9: IQM comparison on `Pong` and `Breakout` after 500K agent steps (*update horizon* = 5, $\lambda = 1$). Our C51-T4 operator achieves clearly higher normalized scores than C51, Rainbow, and DQN, while also exhibiting lower variance. These results indicate that the stochastic truncation mechanism stabilizes multi-step distributional learning even in the early-training regime.

## E  PROOF OF THEOREM 1

We consider the operator $\mathcal{R}$ defined by a clipped importance weight sequence $\{c_t\}$, interpolation factor $\lambda \in [0,1]$, and total variation distance proxy $d$, where we clip the weight between the behavior policy $\mu$ and target policy $\pi$.

Let $Q$ be an arbitrary action-value function and $Q^\pi$ the fixed point of the target Bellman operator $\mathcal{T}^\pi$:

$$\mathcal{T}^\pi Q^\pi = Q^\pi.$$

We define the difference:

$$\Delta Q := Q - Q^\pi.$$

We now derive the deviation of the clipped operator $\mathcal{R}$ from $Q^\pi$:

$$\mathcal{R}Q(s,a) - Q^\pi(s,a) = \sum_{t \geq 1} \gamma^t \mathbb{E}^{a_{1:t}}_{s_{1:t}} \left[ \left( \prod_{i=1}^{t-1} c_i \right) \left( \mathbb{E}_\pi \Delta Q(s_t, \cdot) - c_t \Delta Q(s_t, a_t) \right) \right]$$

$$= \sum_{t \geq 1} \gamma^t \mathbb{E}^{a_{1:t-1}}_{s_{1:t}} \left[ \left( \prod_{i=1}^{t-1} c_i \right) \left( \mathbb{E}_\pi \Delta Q(s_t, \cdot) - \mathbb{E}_{a_t} \left[ c_t(a_t, \mathcal{F}_t) \Delta Q(s_t, a_t) \mid \mathcal{F}_t \right] \right) \right]$$

$$= \sum_{t \geq 1} \gamma^t \mathbb{E}^{a_{1:t-1}}_{s_{1:t}} \left[ \left( \prod_{i=1}^{t-1} c_i \right) \sum_b \left( \pi(b|s_t) - \beta(b|s_t) c_t(b, \mathcal{F}_t) \right) \Delta Q(s_t, b) \right].$$

Let us define weights:

$$w_{y,b} := \sum_{t \geq 1} \gamma^t \mathbb{E}^{a_{1:t-1}}_{s_{1:t}} \left[ \left( \prod_{i=1}^{t-1} c_i \right) \left( \pi(b|s_t) - \beta(b|s_t) c_t(b, \mathcal{F}_t) \right) \mathbf{1}\{s_t = y\} \right].$$

Then the difference becomes:

$$\mathcal{R}Q(s,a) - Q^\pi(s,a) = \sum_{y,b} w_{y,b} \Delta Q(y,b).$$

However, in our setting, sub-convexity does not hold in general due to the possibility of negative weights (when $\pi(b|s_t) < \lambda \beta(b|s_t) d$).

To ensure convergence toward the fixed point $Q^\pi$, we require the operator $\mathcal{R}$ to be a contraction in the supremum norm (also known as $\ell_\infty$-norm). That is, we want the following condition to hold:

$$\|\mathcal{R}Q - \mathcal{R}Q'\|_\infty \leq \kappa \|Q - Q'\|_\infty, \quad \kappa < 1. \tag{18}$$

This ensures that the operator $\mathcal{R}$ brings any two value functions closer under repeated application, eventually converging to a unique fixed point. Our operator deviation is expressed as a weighted combination of differences $\Delta Q(y,b)$, and the contraction factor $\kappa$ can be interpreted as the total weight magnitude:

$$\kappa := \sum_{y,b} |w_{y,b}|, \tag{19}$$

where $w_{y,b}$ is the weight assigned to the deviation term $\Delta Q(y,b)$. Hence, for contraction to hold, we require:

$$\sum_{y,b} |w_{y,b}| < 1. \tag{20}$$

This forms the key criterion for verifying that the operator $\mathcal{R}$ induces a contraction in value space and guarantees convergence under repeated application.

**Deviation Decomposition and Contraction Analysis.** Define, for state $y$ and action $b$,

$$w_{y,b} \;=\; \pi(b \mid y) \;-\; \lambda\,\beta(b \mid y)\,p_t.$$

Then, the total absolute deviation at state $y$ is

$$\sum_b |w_{y,b}| = \sum_{b \in \mathcal{P}} \big(\pi(b \mid y) - \lambda\beta(b \mid y)p_t\big) + \sum_{b \in \mathcal{N}} \big(\lambda\beta(b \mid y)p_t - \pi(b \mid y)\big),$$

where

$$\mathcal{P} = \{b : \pi(b \mid y) \geq \lambda\beta(b \mid y)p_t\}, \qquad \mathcal{N} = \{b : \pi(b \mid y) < \lambda\beta(b \mid y)p_t\}.$$

This simplifies to

$$\sum_b |w_{y,b}| = \sum_b \pi(b \mid y) - \lambda p_t \sum_b \beta(b \mid y) + 2\sum_{b \in \mathcal{N}} \big(\lambda\beta(b \mid y)p_t - \pi(b \mid y)\big) = 1 - \lambda p_t + 2\sum_{b \in \mathcal{N}} \big(\lambda\beta(b \mid y)p_t - \pi(b \mid y)\big).$$

The total weighted deviation over time is

$$\sum_{t \geq 1} \gamma^t \, \mathbb{E}\left[ \xi^{\,t-1} \sum_b \big|\pi(b \mid s_t) - \lambda\beta(b \mid s_t)p_t\big| \right].$$

Using the above decomposition, this equals

$$\sum_{t \geq 1} \gamma^t \, \mathbb{E}\left[ \xi^{\,t-1} \left( 1 - \lambda p_t + 2\sum_{b \in \mathcal{N}_t} \big(\lambda\beta(b \mid s_t)p_t - \pi(b \mid s_t)\big) \right) \right],$$

where $\mathcal{N}_t = \{b : \pi(b \mid s_t) < \lambda\beta(b \mid s_t)p_t\}$.

Thus we obtain the upper bound

$$\|\mathcal{R}Q - Q^\pi\|_\infty \;\leq\; (1 - \lambda p_t) \underbrace{\sum_{t \geq 1} \gamma^t \xi^{\,t-1}}_{= \,\gamma C} \;+\; 2\,\Delta_\lambda,$$

with

$$C := \sum_{t \geq 0} (\gamma\xi)^t = \frac{1}{1 - \gamma\xi}, \qquad \Delta_\lambda := \sum_{t \geq 1} \gamma^t \xi^{\,t-1} \sum_{b \in \mathcal{N}_t} \big(\lambda\beta(b \mid s_t)p_t - \pi(b \mid s_t)\big).$$

**Using $p_t \leq \xi$.** Since $(1 - \lambda p_t)$ decreases in $p_t$, the safe bound is

$$(1 - \lambda p_t)\,\gamma C \;\leq\; \gamma C.$$

Meanwhile,

$$\Delta_\lambda \;\leq\; \lambda \sum_{t \geq 2} \gamma^t \xi^{\,t-1} = \lambda\gamma\xi(C - 1) = \frac{\lambda\gamma^2\xi^2}{1 - \gamma\xi}.$$

Hence the combined bound is

$$\boxed{\|\mathcal{R}Q - Q^\pi\|_\infty \;\leq\; \gamma C + 2\lambda\gamma\xi(C - 1) = \frac{\gamma}{1 - \gamma\xi} + \frac{2\lambda\gamma^2\xi^2}{1 - \gamma\xi}.}$$

**Contraction condition.** For contraction we require

$$\frac{\gamma}{1 - \gamma\xi} + \frac{2\lambda\gamma^2\xi^2}{1 - \gamma\xi} < 1 \quad \Longleftrightarrow \quad \gamma + 2\lambda\gamma^2\xi^2 < 1 - \gamma\xi.$$

Equivalently,

$$\boxed{2\lambda\gamma^2\xi^2 < 1 - \gamma(1 + \xi)}.$$

**Therefore:**

1. **Feasible $\gamma$ (RHS must be positive):**

$$\boxed{0 < \gamma < \frac{1}{1+\xi}}$$

2. **Feasible $\lambda$ for given $\gamma$:**

$$\boxed{0 \leq \lambda \leq \min\left\{1, \frac{1-\gamma(1+\xi)}{2\gamma^2\xi^2}\right\}}, \qquad 0 < \gamma < \frac{1}{1+\xi}$$

3. **Worst case $\xi = 1$:** The global constraint is $\gamma < \frac{1}{2}$. For fixed $\lambda$, the maximal $\gamma$ is

$$\boxed{\gamma_+(\lambda, 1) = \frac{-1 + \sqrt{1+2\lambda}}{2\lambda}} \quad \text{(further capped by } 1/2\text{)}.$$

## F    PROOF OF THEOREM 2

This proof basically follows the same arguments as in the proof of the policy iteration of Retrace (Munos et. al. (2016)) (Munos et al., 2016).

**Step 1. Defining (sub)-probability transition operator**    Since the corresponding probability $p_s$ that $A_s = 1$ is Markovian by the definition in Equation (5), we first examine the follwing expectation:

$$\mathbb{E}_{p_s}\left[\sum_{s'}\sum_{a'}\mathcal{P}(s'|s,a)\beta(a'|s')A_s Q(s',a')\right] = \sum_{s'}\sum_{a'}\mathcal{P}(s'|s,a)\beta(a'|s')p_s(s',a')Q(s',a'). \tag{21}$$

Now, we define the corresponding (sub)-probability transition operator:

$$\sum_{s'}\sum_{a'}\mathcal{P}(s'|s,a)\beta(a'|s')p(s',a')Q(s',a') =: (\mathcal{P}^{p\beta}Q)(s,a). \tag{22}$$

**Step 2. Upper bound on $Q_{n+1} - Q^*$**    We rewrite our T4 operator in Equation (6) as follows.

$$\mathcal{R}_{p,\lambda}Q = Q + \sum_{t\geq 0}(\gamma\lambda)^t(\mathcal{P}^{p\beta})^t(\mathcal{T}^\pi Q - Q) = Q + (I - \gamma\lambda\mathcal{P}^{p\beta})^{-1}(\mathcal{T}^\pi Q - Q) \tag{23}$$

where $(I - \gamma\lambda\mathcal{P}^{p\beta})^{-1} = \sum_{t=0}^{\infty}(\gamma\lambda\mathcal{P}^{p\beta})^t$. Since $Q_{n+1} = \mathcal{R}_{p,\lambda}Q_n$,

$$\begin{aligned}
Q_{n+1} - Q^* &= Q_n - Q^* + (I - \gamma\lambda\mathcal{P}^{p\beta})^{-1}(\mathcal{T}^\pi Q_n - Q_n) \\
&= (I - \gamma\lambda\mathcal{P}^{p\beta})^{-1}[\mathcal{T}^\pi Q_n - Q_n + (I - \gamma\lambda\mathcal{P}^{p\beta})(Q_n - Q^*)] \\
&= (I - \gamma\lambda\mathcal{P}^{p\beta})^{-1}[\mathcal{T}^\pi Q_n - Q^* - \gamma\lambda\mathcal{P}^{p\beta}(Q_n - Q^*)] \\
&= (I - \gamma\lambda\mathcal{P}^{p\beta})^{-1}[\mathcal{T}^\pi Q_n - \mathcal{T}Q^* - \gamma\lambda\mathcal{P}^{p\beta}(Q_n - Q^*)] \\
&\leq (I - \gamma\lambda\mathcal{P}^{p\beta})^{-1}\left[\gamma\lambda\mathcal{P}^\pi(Q_n - Q^*) - \gamma\lambda\mathcal{P}^{p\beta}(Q_n - Q^*)\right] \\
&= \gamma\lambda(I - \gamma\lambda\mathcal{P}^{p\beta})^{-1}[\mathcal{P}^\pi - \mathcal{P}^{p\beta}](Q_n - Q^*) \\
&= \mathcal{B}(Q_n - Q^*),
\end{aligned}$$

where we denote $\gamma\lambda(I - \gamma\lambda\mathcal{P}^{p\beta})^{-1}[\mathcal{P}^\pi - \mathcal{P}^{p\beta}]$ as $\mathcal{B}$. We rewrite $\mathcal{B}$ as

$$\mathcal{B} = \gamma\lambda(I - \gamma\lambda\mathcal{P}^{p\beta})^{-1}[\mathcal{P}^\pi - \mathcal{P}^{p\beta}] = \gamma\lambda\sum_{t\geq 0}(\gamma\lambda)^t(\mathcal{P}^{p\beta})^t(\mathcal{P}^\pi - \mathcal{P}^{p\beta}).$$

To show that $\mathcal{B}$ has non-negative elements, whose sum over each row is at most $\gamma\lambda$. Let $\mathbf{1}$ be the vector with 1-components. We obtain

$$(\mathcal{P}^\pi - \mathcal{P}^{p\beta})\mathbf{1}(s,a) = \sum_{s'}\sum_{a'}\mathcal{P}(s'\mid s,a)\left[\pi(a'\mid s') - p(s',a')\beta(a'\mid s')\right] \geq 0 \qquad (24)$$

Then, we have

$$\mathcal{B}\mathbf{1}(s,a) = \gamma\lambda\sum_{t\geq 0}(\gamma\lambda)^t(\mathcal{P}^{p\beta})^t(P^\pi - \mathcal{P}^{p\beta})\mathbf{1}(s,a)$$

$$= \gamma\lambda\sum_{t\geq 0}(\gamma\lambda)^t(\mathcal{P}^{p\beta})^t\mathbf{1}(s,a) - \sum_{t\geq 0}(\gamma\lambda)^{t+1}(\mathcal{P}^{p\beta})^{t+1}\mathbf{1}(s,a)$$

$$= \mathbf{1}(s,a) - (1-\gamma\lambda)\sum_{t\geq 0}(\gamma\lambda)^t(\mathcal{P}^{p\beta})^t\mathbf{1}(s,a)$$

$$\leq \gamma\lambda\mathbf{1}(s,a).$$

The last inequality is derived by $\sum_{t\geq 0}(\gamma\lambda)^t(\mathcal{P}^{p\beta})^t\mathbf{1} \geq \mathbf{1})$. By the result of Theorem 1, we have

$$Q_{n+1} - Q^* \leq \gamma\lambda\|Q_n - Q^*\|_{p,\infty}\mathbf{1}. \qquad (25)$$

**Step 3. Lower bound on $Q_{n+1} - Q^*$**   By Equation (23), we obtain

$$Q_{n+1} = Q_n + (I - \gamma\lambda\mathcal{P}^{p\beta})^{-1}(\mathcal{T}^\pi Q_n - Q_n)$$

$$= Q_n + \sum_{i\geq 0}(\gamma\lambda\mathcal{P}^{p\beta})^i(\mathcal{T}^\pi Q_n - Q_n)$$

$$= \mathcal{T}^\pi Q_n + \sum_{i\geq 1}(\gamma\lambda\mathcal{P}^{p\beta})^i(\mathcal{T}^\pi Q_n - Q_n)$$

$$= \mathcal{T}^\pi Q_n + \gamma\lambda\mathcal{P}^{p\beta}(I - \gamma\lambda\mathcal{P}^{p\beta})^{-1}(\mathcal{T}^\pi Q_n - Q_n).$$

As we define $\varepsilon_n$ in the statement of Theorem 2, we have

$$\mathcal{T}^{\pi_n}Q_n \geq \mathcal{T}Q_n - \varepsilon_n\|Q_n\| \geq \mathcal{T}^\pi Q_n - \varepsilon_n\|Q_n\|.$$

Then,

$$Q_{n+1} - Q^* = Q_{n+1} - \mathcal{T}^{\pi_n}Q_n + \mathcal{T}^{\pi_n}Q_n - \mathcal{T}^{\pi^*}Q_n + \mathcal{T}^{\pi^*}Q_n - \mathcal{T}^{\pi^*}Q^*$$

$$\geq Q_{n+1} - \mathcal{T}^{\pi_n}Q_n + \gamma\mathcal{P}^{\pi^*}(Q_n - Q^*) - \varepsilon_n\|Q_n\|\mathbf{1}.$$

As a result, we conclude that

$$Q_{n+1} - Q^* \geq \gamma\lambda\mathcal{P}^{p\beta}(I - \gamma\lambda\mathcal{P}^{p\beta})^{-1}(\mathcal{T}^\pi Q_n - Q_n) + \gamma P^\pi(Q_n - Q^*) - \varepsilon_n\|Q_n\|\mathbf{1}. \qquad (26)$$

**Step 4. Lower bound on $\mathcal{T}^\pi Q_n - Q_n$**   Similar to (Munos et al., 2016), we assume that $\varepsilon_n \to 0$, $\mathcal{T}^{\pi^0}Q_0 - Q_0 \geq 0$, and $(\pi_n)$ is increasingly greedy with regard to $(Q_n)$ as follows:

$$\mathcal{T}^{\pi_{n+1}}Q_{n+1} - Q_{n+1} \geq \mathcal{T}^{\pi_n}Q_{n+1} - Q_{n+1}.$$

Let $\mathcal{H}_n = \gamma[P^{\pi_k} - \mathcal{P}^{p\beta}](I - \gamma\lambda\mathcal{P}^{p\beta})^{-1}$. We have

$$\mathcal{T}^{\pi_{n+1}}Q_{n+1} - Q_{n+1} \geq \mathcal{T}^{\pi_n}Q_{n+1} - Q_{n+1}$$

$$= \mathcal{T}^{\pi_n}\mathcal{R}_{p,\lambda}Q_n - \mathcal{R}_{p,\lambda}Q_n$$

$$= r + (\gamma\mathcal{P}^{\pi_n} - I)\mathcal{R}_{p,\lambda}Q_n$$

$$= r + (\gamma\mathcal{P}^{\pi_n} - I)\left[Q_n + (I - \gamma\lambda\mathcal{P}^{p\beta})^{-1}(\mathcal{T}^\pi Q_n - Q_n)\right]$$

$$= \mathcal{T}^{\pi_n}Q_n - Q_n + (\gamma\mathcal{P}^{\pi_n} - I)(I - \gamma\lambda\mathcal{P}^{p\beta})^{-1}(\mathcal{T}^\pi Q_n - Q_n)$$

$$= \gamma[\mathcal{P}^{\pi_k} - \mathcal{P}^{p\beta}](I - \gamma\lambda\mathcal{P}^{p\beta})^{-1}(\mathcal{T}^\pi Q_n - Q_n)$$

$$= \mathcal{H}_n(\mathcal{T}^\pi Q_n - Q_n), \qquad (16)$$

Recall that $\mathcal{P}^{\pi_n} - \mathcal{P}^{p\beta}$ (as shown in Equation (24)) and $(I - \gamma\lambda\mathcal{P}^{p\beta})^{-1}$ have non-negative elements. In the above, we proved that $\mathcal{H}_n$ has non-negative elements as well. Therefore,

$$\mathcal{T}^\pi Q_n - Q_n \geq \mathcal{H}_{n-1}\mathcal{H}_{n-2}\cdots\mathcal{H}_0(\mathcal{T}^{\pi_0}Q_0 - Q_0) \geq 0.$$

Finally, Equation (26) implies that

$$Q_{n+1} - Q^* \geq \gamma\mathcal{P}^{\pi^*}(Q_n - Q^*) - \varepsilon_n\|Q_n\|\mathbf{1}.$$

Combining the above with Equation (25), we have

$$\|Q_{n+1} - Q^*\| \leq \gamma\|Q_n - Q^*\| + \varepsilon_n\|Q_n\|.$$

We note that $Q_n$ is bounded. When $\epsilon_n$ satisfies $\varepsilon_n < (1-\gamma)/2$, we have

$$\|Q_{n+1}\| \leq \|Q^*\| + \gamma\|Q_n - Q^*\| + \frac{1-\gamma}{2}\|Q_n\| \leq (1+\gamma)\|Q^*\| + \frac{1+\gamma}{2}\|Q_n\|.$$

Furthermore,

$$\limsup \|Q_n\| \leq \frac{1+\gamma}{1-(1+\gamma)/2}\|Q^*\|.$$

Since $Q_n$ is bounded, we conclude that $\limsup Q_n = Q^*$.

$\square$

## G    PROOF OF LEMMA 1

We start the proof with the following lemma.

**Lemma 4.** *Under Definition 1, any two stationary distributions $\mu_\beta$ and $\mu_\pi$ of $\mathcal{P}^\beta$ and $\mathcal{P}^\pi$ satisfy $\|\mu_\beta - \mu_\pi\|_{TV} \leq \frac{d}{\alpha}$.*

The proof of Lemmas 4 relies on properties of nearby Markov chains. Detailed proof is provided in Appendix G.1.

**Step 1. Inserting $\mu_\pi$ by triangular inequality**    First, we can upper bound the distance from stationary of $\mathcal{P}^\beta$ by triangular inequality:

$$\left\|\mathcal{P}^{\beta(t)}(s,\cdot) - \mu_\beta\right\|_{\text{TV}} \leq \left\|\mathcal{P}^{\beta(t)}(s,\cdot) - \mu_\pi\right\|_{\text{TV}} + \|\mu_\beta - \mu_\pi\|_{\text{TV}},$$

where $\mu_\pi$ denotes the stationary distribution of $\mathcal{P}^\pi$.

**Step 2. Bounding the distance from stationary**    By using the distance between two nearby Markov chains, we have

$$\left\|\mathcal{P}^{\beta(t)}(s,\cdot) - \mu_\pi\right\|_{\text{TV}} + \|\mu_\beta - \mu_\pi\|_{\text{TV}}$$
$$= \left\|\mathcal{P}^{\beta(t)}(s,\cdot) - \mu_\pi\mathcal{P}^{\pi(t)}\right\|_{\text{TV}} + \|\mu_\beta - \mu_\pi\|_{\text{TV}}$$
$$\leq \max_{s'}\left\|\mathcal{P}^{\beta(t)}(s,\cdot) - \mathcal{P}^{\pi(t)}(s',\cdot)\right\|_{\text{TV}} + \|\mu_\beta - \mu_\pi\|_{\text{TV}}$$
$$\leq \max_{s'}\left\{\left\|\mathcal{P}^{\beta(t)}(s,\cdot) - \mathcal{P}^{\pi(t)}(s,\cdot)\right\|_{\text{TV}} + \left\|\mathcal{P}^{\pi(t)}(s',\cdot) - \mathcal{P}^{\pi(t)}(s,\cdot)\right\|_{\text{TV}}\right\} + \|\mu_\beta - \mu_\pi\|_{\text{TV}}$$
$$\leq \left\|\mathcal{P}^{\beta(t)}(s,\cdot) - \mathcal{P}^{\pi(t)}(s,\cdot)\right\|_{\text{TV}} + \max_{s'}\left\{\left\|\mathcal{P}^{\pi(t)}(s',\cdot) - \mathcal{P}^{\pi(t)}(s,\cdot)\right\|_{\text{TV}}\right\} + \|\mu_\beta - \mu_\pi\|_{\text{TV}}$$

By Lemma 4, we have

$$\left\|\mathcal{P}^{\beta^{(t)}}(s,\cdot) - \mathcal{P}^{\pi^{(t)}}(s,\cdot)\right\|_{\mathrm{TV}} + \max_{s'}\left\|\mathcal{P}^{\pi^{(t)}}(s,\cdot) - \mathcal{P}^{\pi^{(t)}}(s',\cdot)\right\|_{\mathrm{TV}} + \|\mu_\beta - \mu_\pi\|_{\mathrm{TV}}$$

$$\leq \left\|\mathcal{P}^{\beta^{(t)}}(s,\cdot) - \mathcal{P}^{\pi^{(t)}}(s,\cdot)\right\|_{\mathrm{TV}} + (1-\alpha)^t + \frac{d}{\alpha}$$

$$\leq \left\|\mathcal{P}^{\beta^{(t)}}(s,\cdot) - \mathcal{P}^{\pi^{(t)}}(s,\cdot)\right\|_{\mathrm{TV}} + 1 - \alpha + \frac{d}{\alpha}.$$

$\square$

## G.1 Proof of Lemma 4

By the triangle inequality,

$$\|\mu_\beta - \mu_\pi\|_{\mathrm{TV}} \leq \left\|\mu_\beta \mathcal{P}^\beta - \mu_\pi \mathcal{P}^\beta\right\|_{\mathrm{TV}} + \left\|\mu_\pi \mathcal{P}^\beta - \mu_\pi \mathcal{P}^\pi\right\|_{\mathrm{TV}}$$
$$= (1-\alpha)\|\mu_\beta - \mu_\pi\|_{\mathrm{TV}} + d.$$

Each term in the second line is derived from the ergodicity of Markov chain and Definition 1, respectively. Then, we have

$$\|\mu_\beta - \mu_\pi\|_{\mathrm{TV}} \leq \frac{d}{\alpha}.$$

$\square$

## H Proof of Lemma 2

The proof is basically the same as Theorem 9 in (Johndrow & Mattingly, 2017a) with minor modification. We construct a coupling $(S_t^\beta, S_t^\pi)$ to examine the long-time dynamic of the agreement between $S_t^\beta$ and $S_t^\pi$.

**Step 1. Construction of the Coupling** Given any two probability measures $m_1$ and $m_2$ on $\mathcal{S}$, it is known that

$$\|m_1 - m_2\|_{\mathrm{TV}} = 1 - \min(m_1, m_2)(\mathcal{S}) = [m_1 - m_2]^+(\mathcal{S}) = [m_2 - m_1]^+(\mathcal{S}).$$

Now we compare two transitions $\mathcal{P}^\beta$ and $\mathcal{P}^\pi$ where the transition kernel $\mathcal{P}$ is uniformly $d$-bounded. For any $\xi = (\xi_1, \xi_2) \in \mathcal{S} \times \mathcal{S}$, we define the measures on $\mathcal{S}$

$$Q_d(\xi, \cdot) = \frac{\min(\mathcal{P}^\pi(\xi_1, \cdot), \mathcal{P}^\beta(\xi_2, \cdot))}{\rho_d(\xi)}, \quad R_d(\xi, \cdot) = \frac{[\mathcal{P}^\pi(\xi_1, \cdot) - \mathcal{P}^\beta(\xi_2, \cdot)]^+}{1 - \rho_d(\xi)},$$

$$\widetilde{R}_d(\xi, \cdot) = \frac{[\mathcal{P}^\beta(\xi_2, \cdot) - \mathcal{P}^\pi(\xi_1, \cdot)]^+}{1 - \rho_d(\xi)},$$

where $\rho_d(\xi)$ denotes

$$\rho_d(\xi) = 1 - \|\mathcal{P}^\pi(\xi_1, \cdot) - \mathcal{P}^\beta(\xi_2, \cdot)\|_{\mathrm{TV}}.$$

We note that these three measures are all probability measures on $\mathcal{S}$ for fixed $\xi \in \mathcal{S} \times \mathcal{S}$. Now we define the transition kernels in $\mathcal{S} \times \mathcal{S}$ for $\xi = (\xi_1, \xi_2)$ and $s = (s_1, s_2)$ in $\mathcal{S} \times \mathcal{S}$:

$$Q_d(\xi, ds) = \rho_d(\xi)Q_d(\xi, ds_1)\delta_{s_1}(ds_2) + (1 - \rho_d(\xi))\left(R_d(\xi, ds_1) \times \widetilde{R}_d(\xi, ds_2)\right). \quad (27)$$

**Step 2. Using stochastic dominance among random variables** For the following derivations, we first define a stochastic process $Z_n^d$

$$Z_n^d = \begin{cases} 0 & \text{if } S_n^\beta = S_n^\pi \\ 1 & \text{if } S_n^\beta \neq S_n^\pi \end{cases} \quad (28)$$

where $(S_t^\beta, S_t^\pi)$ be the Markov chain on $\mathcal{S} \times \mathcal{S}$ defined by the above transition density $Q_d$ in Equation (27). Since $Z_n^d$ is not Markovian, we define the probability $\mathrm{P}(Z_{n+1}^d = k \mid Z_n^d = j)$ as $\mathbb{E}[\mathbf{1}\{Z_{n+1}^d = k\} \mid Z_n^d = j]$. **Note that $Z_k^d$ corresponds to the Bernoulli indicator $A_k$.** Now we have

$$\mathrm{P}(Z_{n+1}^d = 0 \mid Z_n^d = 0) \geq 1 - d \quad \text{and} \quad \mathrm{P}(Z_{n+1}^d = 0 \mid Z_n^d = 1) \geq \rho$$

with probability 1. Let $Y_n$ be the Markov chain on $\{0, 1\}$ with the transition matrix

$$P_d = \begin{pmatrix} 1 - d & d \\ \rho & 1 - \rho \end{pmatrix} \tag{29}$$

and assume that $d < 1 - \rho$. We have

$$\mathrm{P}(Z_{n+1}^d = 0 \mid Z_n^d = 0) \geq \mathrm{P}(Y_{n+1} = 0 \mid Y_n = 0) = 1 - d,$$
$$\mathrm{P}(Z_{n+1}^d = 0 \mid Z_n^d = 1) \geq \mathrm{P}(Y_{n+1} = 0 \mid Y_n = 1) = \rho,$$
$$\mathrm{P}(Z_{n+1}^d = 0 \mid Z_n^d = 0) \geq \mathrm{P}(Y_{n+1} = 0 \mid Y_n = 1) = \rho.$$

with probability 1. This result implies that

$$\mathrm{P}(Z_{n+1}^d \leq k \mid Z_n^d \leq Y_n) \geq \mathrm{P}(Y_{n+1} \leq k \mid Z_n^d \leq Y_n) \tag{30}$$

for all $k \geq 0$ and $n \geq 0$. It is equivalent to the definition of stochastic dominance, then we can construct a monotone coupling of the processes $Y_n$ and $Z_n^d$ where

$$\mathrm{P}(Z_n^\epsilon \leq Y_n \text{ for all } n) = 1 \tag{31}$$

and $Z_0^d \leq Y_0$. Finally, with probability 1, we have

$$\frac{1}{n} \sum_{k=0}^{n-1} \mathbf{1}\{S_k^\beta \neq S_k^\pi\} = \frac{1}{n} \sum_{k=0}^{n-1} \mathbf{1}\{Z_k^d = 1\} \leq \frac{1}{n} \sum_{k=0}^{n-1} \mathbf{1}\{Y_k = 1\}. \tag{32}$$

We note that it is enough to bound the amount of time $Y_n = 1$ to control the fraction of the time that $S_n^\beta$ and $S_n^\pi$ disagree.

**Step 3. Bounding chain in expectation**   The key idea in our proof is to leverage the fact that $Z_n^d$ is stochastically dominated by $Y_n$. By explicitly analyzing the amount of time that $Y_n$ spends in state 1, we can derive bounds relevant to the problems of interest. Let a Markov transition matrix of the bounding chain be

$$\mathcal{P}_d = \begin{pmatrix} 1 - d & d \\ \rho & 1 - \rho \end{pmatrix}. \tag{33}$$

We know that the Markov chain $\mathcal{P}_d$ has a generator $L_d = \mathcal{P}_d - I$ and its unique stationary measure $\mu_d$ denoted by

$$\mu_d = \left( \frac{\rho}{\rho + d}, \frac{d}{\rho + d} \right). \tag{34}$$

Note that, by definition, $\mu_d L_d = 0$ and $\mu_d \mathcal{P}_d = \mu_d$. We define the following vectors

$$\phi = \begin{pmatrix} 0 \\ 1 \end{pmatrix}, \quad \mathbf{1} = \begin{pmatrix} 1 \\ 1 \end{pmatrix}, \quad \bar{\phi}_d = \mu_d \phi \mathbf{1} = \left( \frac{d}{\rho + d}, \frac{\rho}{\rho + d} \right), \quad \text{and} \quad \tilde{\phi}_d = \phi - \bar{\phi}_d = \left( \frac{-d}{\rho + d}, \frac{\rho}{\rho + d} \right).$$

Let $\psi_d$ be the solution to the following equation

$$L_d \psi_d = -\tilde{\phi}_d. \tag{35}$$

Then, we can easily see that

$$\psi_d = \sum_{k=0}^{\infty} \mathcal{P}_d^{(k)} \tilde{\phi}_d. \tag{36}$$

Consider $w_d = \begin{pmatrix} -\epsilon \\ 1 \end{pmatrix}$. It satisfies $\mathcal{P}_d w_d = (1 - \rho - d)w_d$, then $w_d$ is a right-eigenvector with eigenvalue $1 - \rho - d$. Since $\tilde{\phi}_d = \frac{\rho}{\rho+d} w_d$, we have

$$\psi_d = \left( \frac{\rho}{\rho + d} \right) \left( \sum_{k=0}^{\infty} (1 - \rho - d)^k \right) w_d = \frac{\rho}{(\rho + d)^2} w_d.$$

We note that $d < 1 - \rho$ by definition so that $1 - \rho - d \in (0, 1)$. For any initial distribution of $(S_1^\beta, S_1^\pi)$ induced by $\mathcal{P}^\beta \mathcal{P}_0$ and $\mathcal{P}^\pi \mathcal{P}_0$, we define the initial distribution of $Y_n$ as

$$\nu(0) = \mathrm{P}(S_1^\beta = S_1^\pi) \text{ and } \nu(1) = \mathrm{P}(S_1^\beta \neq S_1^\pi),$$

respectively. Combining the above properties, we have

$$\nu \mathcal{P}_d^n \psi_d - \nu \psi_d = \sum_{k=0}^{n-1} \nu \mathcal{P}_d^k L_d \psi_d = \sum_{k=0}^{n-1} \nu \mathcal{P}_d^k \phi - n\nu\bar{\phi}_d.$$

Rearranging the above equation, we finally have

$$\frac{1}{n} \sum_{k=0}^{n-1} \nu \mathcal{P}_d^k \phi = \frac{d}{\rho + d} + \frac{\nu \mathcal{P}_d^n \psi_d \nu \psi_d}{n} = \frac{d}{\rho + d} + \frac{\rho}{n(\rho + d)^2} (1 - (1 - \rho - d)^n)\nu w_d$$

$$= \frac{d}{\rho + d} + \frac{1 - (1 - \rho - d)^n}{n(\rho + d)^2} (\rho \mathrm{P}(S_1^\beta \neq S_1^\pi) - d(1 - \mathrm{P}((S_1^\beta \neq S_1^\pi))))$$

$$= \frac{d}{\rho + d} + \frac{1 - (1 - \rho - d)^n}{n(\rho + d)} (\mathrm{P}(S_1^\beta \neq S_1^\pi) - \frac{d}{\rho + d}).$$

Note that

$$\frac{1}{n} \sum_{k=0}^{n-1} \mathbf{1}\{Y_k = 1\} = \frac{1}{n} \sum_{k=0}^{n-1} \phi(Y_k)$$

by the definition in Equation (32). We conclude that

$$\frac{1}{n} \sum_{k=1}^{n} \mathrm{P}(S_k^\beta \neq S_k^\pi) \leq \frac{1}{n} \sum_{k=0}^{n-1} \nu \mathcal{P}_d^k \phi = \frac{d}{\rho + d} + \frac{1 - (1 - \rho - d)^n}{n(\rho + d)} \left( \mathrm{P}(S_1^\beta \neq S_1^\pi) - \frac{d}{\rho + d} \right). \quad (37)$$

$\square$

## I  PROOF OF THEOREM 3

**Step 1. Analyze the property of coupling time**  To begin with, recall that the Markov chain $Y_n$ on $\{0, 1\}$ has the following transition matrix

$$\mathcal{P}_d = \begin{pmatrix} 1 - d & d \\ \rho & 1 - \rho \end{pmatrix}.$$

We introduce the conditional expectation of a random variable with respect to $\sigma$-algebra. Define a filtration $\mathcal{F}_n = \sigma(Y_0, Y_1, Y_2, \cdots, Y_n)$ and $\{Y_n\}$ is adapted to $\{\mathcal{F}_n\}$ and a stopping time

$$\tau_d = \min\{n \geq 0 \mid Y_n = 0\}.$$

Since $T_{\beta,\pi} = \min\{t \geq 1 : S_n^\beta = S_n^\pi \mid S_0 \sim \mathcal{P}_0\}$ is the first meeting time of two processes $(S_n^\beta, S_n^\pi)$ defined in Lemma 2, we can rewrite $T_{\beta,\pi}$ as

$$T_{\beta,\pi} = \min\{t \geq 1 : S_n^\beta = S_n^\pi \mid S_0 \sim \mathcal{P}_0\} = \min\{t \geq 1 : Z_n^d = 0\}.$$

Suppose that $\mathbb{E}[\tau] < \infty$. We first note that the stochastic ordering $T_{\beta,\pi} \geq \tau_d$ since we construct a monotone coupling of $Y_n$ and $Z_n^d$ in Equation (30). Then, for any meeting time $\tau$, we have

$$\mathrm{P}(T_{\beta,\pi} \leq \tau) \leq \mathrm{P}(\tau_d \leq \tau).$$

Therefore,

$$P(\tau_d > \tau) = \sum_{k=1}^{\infty} P(\tau_d > k)P(\tau = k) = \sum_{j=1}^{\infty} (1 - \rho)^{j-1}P(\tau = j) = \mathbb{E}[(1 - \rho)^{\tau}]. \quad (38)$$

Let $\Lambda(\rho) = \mathbb{E}[(1 - \rho)^{\tau}]$. To obtain an intuitive result, assume that $\tau \leq N$ almost surely for some constant $N$. Under this assumption, $\Lambda(\rho)$ is differentiable everywhere. In the region $\rho \in (0, \epsilon_0)$ for some small $\epsilon_0 > 0$, the Lagrange remainder of its Taylor expansion yields:

$$\Lambda(\rho) = 1 - \rho\mathbb{E}[\tau] + \frac{1}{2}\Lambda''(\xi)\mathbb{E}[\tau(\tau - 1)] \geq 1 - \rho\mathbb{E}[\tau]$$

for some $\xi \in (0, \rho)$.

Finally, we have

$$P(\tau_d > \tau) \geq 1 - \rho\mathbb{E}[\tau]. \quad (39)$$

Therefore,

$$P(T_{\beta,\pi} \leq \tau) \leq P(\tau_d \leq \tau) \leq \rho\mathbb{E}[\tau]. \quad (40)$$

We conclude that the probability of $T_{\beta,\pi}$ is well defined and note that its value is upper-bounded by the chain $Y_n$. In the next step, we evaluate the mixing time by using the practical stopping time that two process first meet.

**Step 2. Upper bound of the distance from the stationary of $\mathcal{P}^{\beta}$** By Lemma 1, we can upper bound

$$\left\|\mathcal{P}^{\beta(t)}(s, \cdot) - \mu_{\beta}\right\|_{\text{TV}} \leq \left\|\mathcal{P}^{\beta(t)}(s, \cdot) - \mathcal{P}^{\pi(t)}(s, \cdot)\right\|_{\text{TV}} + 1 - \alpha + \frac{d}{\alpha}. \quad (41)$$

By the property of coupling, we have

$$\left\|\mathcal{P}^{\beta(t)}(s, \cdot) - \mu_{\beta}\right\|_{\text{TV}} \leq \left\|\mathcal{P}^{\beta(t)}(s, \cdot) - \mathcal{P}^{\pi(t)}(s, \cdot)\right\|_{\text{TV}} + 1 - \alpha + \frac{d}{\alpha}$$

$$\leq P(S_t^{\beta} \neq S_t^{\pi}|S_0^{\beta} = s, S_0^{\pi} = s) + 1 - \alpha + \frac{d}{\alpha}$$

$$\leq P(T_{\beta,\pi} > t|S_0^{\beta} = s, S_0^{\pi} = s) + 1 - \alpha + \frac{d}{\alpha}.$$

Applying Markov's inequality, we have

$$\|\mathcal{P}^{\beta(t)}(s, \cdot) - \mu_{\beta}\|_{\text{TV}} \leq P(T_{\beta,\pi} > t|S_0^{\beta} = s, S_0^{\pi} = s) + 1 - \alpha + \frac{d}{\alpha}$$

$$\leq \frac{\mathbb{E}[T_{\beta,\pi}]}{t} + (1 - \alpha + \frac{d}{\alpha})$$

To evaluate the mixing time of $\mathcal{P}^{\beta}$, we approximate the above inequality as

$$\|\mathcal{P}^{\beta(t)}(s, \cdot) - \mu_{\beta}\|_{\text{TV}} \leq P(T_{\beta,\pi} > t|S_0^{\beta} = s, S_0^{\pi} = s) + 1 - \alpha + \frac{d}{\alpha}$$

$$\lesssim \frac{\mathbb{E}[T_{\beta,\pi}]}{t}.$$

Then, the mixing time of $\mathcal{P}^{\beta}$, $\tau_{\text{mix}}$ is derived as

$$\tau_{\text{mix}} \leq 2e\mathbb{E}[T_{\beta,\pi}] \quad (42)$$

We now use the estimate of the mixing time $\tau_{\text{mix}}$ as $\mathbb{E}[T_{\beta,\pi}]$ approximately. Then, we have the following result by Equation (4):

$$K = \min(\frac{1}{1 - \gamma}, \mathbb{E}[T_{\beta,\pi}]). \quad (43)$$

$\square$

## J   JUSTIFICATION OF THE ACTION-LEVEL APPROXIMATION FOR THE MIXING-TIME SURROGATE

In this section, we formally justify why our implementation uses an action-level overlap proxy $\hat{p}_i = 1 - \min\{\beta(a_i \mid s_i), \pi(a_i \mid s_i)\}$ to approximate the theoretical state-transition discrepancy $p_i = \mathrm{TV}(P_\beta(\cdot \mid s_i), P_\pi(\cdot \mid s_i))$, and why this is mathematically consistent under the structure of Markov decision processes.

### J.1   THREE LEVELS OF KERNELS

Let $P_{\mathrm{env}}(s' \mid s, a)$ denote the environment transition kernel, which is independent of the policy. Given policies $\beta$ and $\pi$, the induced state-transition kernels are

$$P_\beta(s' \mid s) = \sum_a \beta(a \mid s) \, P_{\mathrm{env}}(s' \mid s, a), \tag{44}$$

$$P_\pi(s' \mid s) = \sum_a \pi(a \mid s) \, P_{\mathrm{env}}(s' \mid s, a). \tag{45}$$

When we consider the augmented Markov chain on state–action pairs $(S_t, A_t)$, the corresponding kernels are

$$Q_\beta((s, a), (s', a')) = P_{\mathrm{env}}(s' \mid s, a) \, \beta(a' \mid s'), \tag{46}$$

$$Q_\pi((s, a), (s', a')) = P_{\mathrm{env}}(s' \mid s, a) \, \pi(a' \mid s'). \tag{47}$$

Our off-policy data consist of trajectories of the form $(s_t, a_t, s_{t+1})$, which are exact samples from the chain $Q_\beta$.

Following Duan et al. (2021), the multi-step TD error trade-off depends on the mixing properties of the underlying Markov chain. In off-policy evaluation, we are interested in mixing properties involving both $P_\beta$ and a nearby chain $P_\pi$.

### J.2   FROM POLICIES TO STATE-TRANSITION KERNELS

A key structural fact is that the environment transition kernel is policy-independent. Therefore, for each state $s$,

$$P_\beta(\cdot \mid s) = \beta(\cdot \mid s) \, K_s, \qquad P_\pi(\cdot \mid s) = \pi(\cdot \mid s) \, K_s, \tag{48}$$

where the stochastic kernel $K_s$ maps actions to next states:

$$K_s(s' \mid a) \ := \ P_{\mathrm{env}}(s' \mid s, a). \tag{49}$$

Thus, $P_\beta(\cdot \mid s)$ and $P_\pi(\cdot \mid s)$ arise from pushing different action distributions through the same transition map.

### J.3   DATA-PROCESSING INEQUALITY FOR TV DISTANCE

Define the action-level and state-level total variation distances:

$$\mathrm{TV}_{\mathrm{action}}(s) := \frac{1}{2} \sum_a |\beta(a \mid s) - \pi(a \mid s)| = 1 - \sum_a \min\{\beta(a \mid s), \, \pi(a \mid s)\}, \tag{50}$$

$$\mathrm{TV}_{\mathrm{state}}(s) := \frac{1}{2} \sum_{s'} |P_\beta(s' \mid s) - P_\pi(s' \mid s)| = p_i. \tag{51}$$

Since a stochastic kernel cannot increase total variation distance (data-processing inequality),

$$\mathrm{TV}_{\mathrm{state}}(s) \ \leq \ \mathrm{TV}_{\mathrm{action}}(s) \qquad \text{for all } s. \tag{52}$$

This property will be formalized and elaborated in Section J.6 when we introduce the coupling-based relation between the two transition kernels. Hence, $\mathrm{TV}_{\mathrm{action}}(s)$ is an upper bound on the theoretical quantity $p_i$ that governs meeting-time and mixing-time behavior in our coupling analysis.

## J.4 SAMPLE-BASED APPROXIMATION ALONG $\beta$-TRAJECTORIES

The exact action-level TV requires evaluating $\sum_a \min\{\beta(a \mid s_i), \pi(a \mid s_i)\}$. In practice, along a trajectory generated by $\beta$, we only observe a single action sample $a_i \sim \beta(\cdot \mid s_i)$. We therefore adopt the stochastic proxy

$$\hat{p}_i = 1 - \min\{\beta(a_i \mid s_i), \pi(a_i \mid s_i)\}. \tag{53}$$

This is a noisy sample-level approximation of $\mathrm{TV}_{\mathrm{action}}(s_i)$.

While $\hat{p}_i$ is not an unbiased estimator of $p_i$ or of $\mathrm{TV}_{\mathrm{action}}(s_i)$, it preserves the key monotonic relationship: larger policy mismatch leads to larger $\hat{p}_i$, which in turn produces shorter expected truncation lengths in our T4 mechanism. Importantly, $\hat{p}_i$ is fully model-free: it can be computed without access to the environment transition kernel.

## J.5 IMPLICATIONS FOR MIXING-TIME SURROGATES

Our theoretical analysis uses the state-transition TV distance $p_i = \mathrm{TV}(P_\beta(\cdot \mid s_i), P_\pi(\cdot \mid s_i))$ as the quantity governing the meeting time between the two chains. By data processing, this value is upper bounded by the action-level TV. Our implementation uses the sample-based surrogate $\hat{p}_i$, which approximates this action-level quantity.

Thus, the use of $\hat{p}_i$ is mathematically consistent: it provides a directional, model-free proxy for the theoretical $p_i$ and retains its qualitative dependence on policy discrepancy, allowing us to translate mixing-time insights (as in Duan et al. 2021) to the off-policy setting.

**Remark.** A practical consequence of using the approximate disagreement proxy $\hat{p}_i = 1 - \min\{\beta(a_i \mid s_i), \pi(a_i \mid s_i)\}$ is that we effectively rely on an estimated total variation distance that upper bounds the true value. Formally, the stochastic mapping $(s, a) \mapsto s'$ satisfies the data-processing inequality, which implies

$$\mathrm{TV}(P_\beta(\cdot \mid s_i), P_\pi(\cdot \mid s_i)) \leq \mathrm{TV}(\beta(\cdot \mid s_i), \pi(\cdot \mid s_i)) \leq \hat{p}_i.$$

Hence, in the context of Definition 1, the practical estimator corresponds to using a constant $d' \geq d$.

As illustrated in Figure 3b, increasing $d$ slightly enlarges the upper bound on the time-averaged disagreement,

$$\frac{1}{t}\sum_{k=1}^{t} \mathbb{E}[A_k],$$

which in turn increases the resulting first meeting time estimate $\hat{T}_{\beta,\pi}$. Consequently, the truncation length is determined using an estimate that is greater than or equal to the ground-truth meeting time. This does not invalidate our theoretical guarantees, because the condition in Theorem 2 requires only that

$$K \geq \min\left((1-\gamma)^{-1}, \; \mathbb{E}[T_{\beta,\pi}]\right),$$

and any overestimation of $\mathbb{E}[T_{\beta,\pi}]$ still preserves this requirement. Therefore, the main claims of the paper remain valid even when using the practical estimator.

## J.6 DATA-PROCESSING INEQUALITY FOR TOTAL VARIATION DISTANCE

**Lemma (TV contraction under Markov kernel).** Let $P$ and $Q$ be two probability measures over a measurable space $(\mathcal{X}, \mathcal{F})$, and let $f : \mathcal{X} \to \mathcal{Y}$ be a stochastic map, i.e., a Markov kernel such that $f(\cdot \mid x)$ is a probability distribution over $\mathcal{Y}$ for each $x \in \mathcal{X}$. Then the push-forward measures $P_Y, Q_Y$ on $\mathcal{Y}$ defined by

$$P_Y(B) = \int_{\mathcal{X}} f(B \mid x)\, dP(x), \quad Q_Y(B) = \int_{\mathcal{X}} f(B \mid x)\, dQ(x)$$

satisfy the total variation contraction inequality:

$$\mathrm{TV}(P_Y, Q_Y) \leq \mathrm{TV}(P, Q).$$

**Proof.** By definition of total variation distance and linearity of integration, we have:

$$\mathrm{TV}(P_Y, Q_Y) = \sup_{B \subset \mathcal{Y}} |P_Y(B) - Q_Y(B)|$$

$$= \sup_{B \subset \mathcal{Y}} \left| \int_{\mathcal{X}} f(B \mid x)\, dP(x) - \int_{\mathcal{X}} f(B \mid x)\, dQ(x) \right|$$

$$= \sup_{B} \left| \int_{\mathcal{X}} f(B \mid x)\, (dP(x) - dQ(x)) \right|$$

$$\leq \int_{\mathcal{X}} \sup_{B} |f(B \mid x)|\; |dP - dQ|(x)$$

$$\leq \int_{\mathcal{X}} |dP - dQ|(x) = 2 \cdot \mathrm{TV}(P, Q).$$

Using the fact that $\mathrm{TV}(P, Q) = \frac{1}{2} \int |dP - dQ|$, we conclude:

$$\boxed{\mathrm{TV}(P_Y, Q_Y) \leq \mathrm{TV}(P, Q)} \quad \blacksquare$$

**Interpretation.** This result formalizes the intuition that applying a stochastic transformation (Markov kernel) can only reduce, not increase, the distinguishability of two distributions. In reinforcement learning, this inequality explains why the discrepancy between next-state distributions under two policies is always upper bounded by their action-level difference:

$$\mathrm{TV}\left( \sum_a \pi(a \mid s)\mathcal{P}(\cdot \mid s, a), \sum_a \beta(a \mid s)\mathcal{P}(\cdot \mid s, a) \right) \leq \mathrm{TV}(\pi(\cdot \mid s), \beta(\cdot \mid s)).$$

This principle underlies the use of disagreement probabilities $p_i$ in stochastic truncation (T4) as a proxy for policy divergence propagated through the dynamics.

