# OpenReview forum: "Stochastic Truncation for Multi-Step Off-Policy RL"
_ICLR.cc/2026/Conference — Submitted to ICLR 2026_

### Official Review · Reviewer_uLFM · 2025-10-18

**Soundness:** 3
**Presentation:** 2
**Contribution:** 2
**Rating:** 4
**Confidence:** 4

**Summary:**

This paper addresses key challenges in multi-step off-policy RL: distribution mismatch between behavior/target policies amplifies error with longer horizons, existing conservative (e.g., Retrace) and non-conservative (e.g., Peng’s \(Q(\lambda)\)) methods face trade-offs between convergence and variance, and truncation horizon selection (tied to Markov chain mixing time) is hard to estimate online.

The authors proposes T4, a stochastic adaptive truncation mechanism under Retrace. T4 uses coupling analysis to derive a mixing time upper bound, employs Bernoulli variables to truncate trajectories at the first state match of behavior/target policies, and is non-conservative yet convergent under arbitrary behavior policies, with robustness to cap length tuning.

Empirically, T4 (with SAC/TD3) outperforms baselines (one-step, uncorrected n-step, Retrace, Peng’s \(Q(\lambda)\)) in convergence and performance on 5 MuJoCo tasks, and excels in sparse-reward scenarios and matches model-based MBPO’s sample efficiency.

**Strengths:**

It addresses the core issue of multi-step off-policy RL (truncation horizon selection) by proposing T4, an adaptive stochastic truncation mechanism guided by mixing time upper bounds, avoiding the limitations of fixed cap lengths.

T4 achieves a theoretical breakthrough: it is non-conservative (unlike Retrace with high variance) yet guarantees convergence under arbitrary behavior policies (solving the defect of non-conservative methods like Peng’s Q(λ) lacking guarantees)

**Weaknesses:**

1, The paper fails to clarify the necessity of proposing T4, as it does not analyze the limitations of mature off-policy methods (e.g., GTD2, TDC, ETD,Truncated ETD) in multi-step scenarios or explain T4’s advantages over them

2, It relies on strict, impractical assumptions (uniformly d-bounded kernel, cross-Doeblin condition) for T4’s convergence derivation, with no guidance on parameter tuning in practice

3, Experiments are only conducted on 5 MuJoCo continuous control tasks, lacking validation in discrete-action or sparse-reward environments, and counterexamples such as 2 states counterexample, Baird's counterexample.

4, Baseline comparisons are incomplete—only 4 methods (One-step RL, Retrace, etc.) are included, excluding state-of-the-art off-policy algorithms

**Questions:**

1, The paper fails to clarify the necessity of proposing T4, as it does not analyze the limitations of mature off-policy methods (e.g., GTD2, TDC, ETD,Truncated ETD) in multi-step scenarios or explain T4’s advantages over them

2, It relies on strict, impractical assumptions (uniformly d-bounded kernel, cross-Doeblin condition) for T4’s convergence derivation, with no guidance on parameter tuning in practice

3, Experiments are only conducted on 5 MuJoCo continuous control tasks, lacking validation in discrete-action or sparse-reward environments, and counterexamples such as 2 states counterexample, Baird's counterexample.

4, Baseline comparisons are incomplete—only 4 methods (One-step RL, Retrace, etc.) are included, excluding state-of-the-art off-policy algorithms

---

> ### Author Response · Authors · 2025-11-30
>
> We appreciate the reviewer’s thoughtful comments. Several concerns, however, seem to arise from a mismatch in problem setting or from interpreting theoretical assumptions as practical algorithmic requirements. We clarify these misunderstandings below and explain why our assumptions and baseline choices are appropriate within the standard multi-step off-policy framework studied in this work.
>
> ## (W1, W4, Q1, Q4) the comparison with other off-policy methods, e.g. GTD2, TDC, ETD,Truncated ETD
>
> We interpret the reviewer’s references as GTD2 [2], TDC [3], ETD [4], and Truncated ETD [5], all of which are summarized in Sutton & Barto [1] and belong to the class of *off-policy TD methods with linear function approximation*. Among these, [2–4] are fundamentally **one-step TD algorithms**, while [5] introduces an eligibility-trace variant aimed at mitigating the variance of ETD rather than extending the multi-step return horizon. If the reviewer intended different algorithms, clarification would be welcome.
>
> These methods were developed to stabilize the convergence of gradient-based TD under off-policy sampling. Although they involve importance ratios, they focus on correcting semi-gradient TD instability and were originally formulated for **linear function approximation**. Importantly, they do **not** address multi-step off-policy mismatch across horizons or the bias–variance issues inherent in multi-step bootstrapping.
>
> For these reasons, we do not view [2–5] as appropriate baselines for T4. Applying them to deep RL or multi-step tabular operators would require substantial additional algorithmic development and addresses a problem orthogonal to the one studied here.
>
> Among recent methods, the closest multi-step variant applicable to deep RL is the $\alpha$-trace of Rowland et al. [6]. Preliminary tests in continuous-control domains indicated that it does not perform competitively; hence it was not included in the main results. We will nevertheless include its performance in the additional discrete-action experiments that will be uploaded.
>
> [1] Sutton, R. S., & Barto, A. G. (2018). *Reinforcement Learning: An Introduction* (2nd ed.). MIT Press.
>
> [2]  Maei, H.R. and Sutton, R.S. (2010). *GQ(λ): A general gradient algorithm for temporal-difference prediction learning with eligibility traces.*
>
> [3] Sutton, R.S. et al. (2009). *Fast gradient-descent methods for temporal-difference learning with linear function approximation.* ICML.
>
> [4] Sutton, R.S., Mahmood, A.R., and White, M. (2016). *An emphatic approach to the problem of off-policy temporal-difference learning.* JMLR.
>
> [5] Yu, H. (2015). *On convergence of emphatic temporal-difference learning.* COLT.
>
> [6] Rowland, Mark, Will Dabney, and Rémi Munos. "Adaptive trade-offs in off-policy learning." *International Conference on Artificial Intelligence and Statistics*. PMLR, 2020.
>
> ## (W2, Q2) on the “strict, impractical assumptions”
>
> The mixing-time and nearby-kernel assumptions used in our analysis serve solely as **analytical devices** and do not affect the practical operation of T4, which uses only policy-overlap–based disagreement estimates. We expand on this below.
>
> ### Empirical validity of the assumptions
>
> Even in settings with fragile kernel overlap, such as the CliffWalking environment, Figures 2 and 3(a) (newly added in the revision) show that Definition 1 (uniform $d$-bounded kernels) and Assumption 1 (cross-Doeblin) behave in a controlled manner. CliffWalking contains **absorbing states** and **irreversible transitions**, which naturally induce:
>
> - **large kernel gaps**, with
>
> $$
> d(s)=\left\| P^{\beta}(s,\cdot) - P^{\pi}(s,\cdot) \right\|_{\mathrm{TV}}\approx 1
> $$
>
> near the cliff;
>
> - **tiny overlap**, with
>
> $$
> \rho(s,s')=\sum_{x \in \mathcal{S}}\min (P^{\beta}(s,x), P^{\pi}(s',x) ) \approx0
> $$
>
> Despite this worst-case structure—which aligns with the reviewer’s concern—meeting-time–based truncation remains meaningful. Figure 3(b) shows that even when $d$ is large and $\rho$ is small:
>
> As shown in Figure 3(b), even for relatively large $d$ and small $\rho$:
>
> - the empirical $d$ never approaches $1$ (max $\approx 0.7$);
> - $\rho$ is small but consistently positive;
> - and, importantly, the RHS of Lemma 2 falls below $1$ for all $(d,\rho)$ once $t \ge 10$.
>
> Together, these observations indicate that
>
> $$
> \Pr(S_t^\beta = S_t^\pi) > 0 \quad \text{for all tested regimes},
> $$
>
> ensuring that the first meeting time is finite and that T4’s truncation remains well-defined and practically meaningful. The assumptions therefore do not require $P_\beta$ and $P_\pi$ to be “close,” but only that their discrepancy does not prevent eventual coalescence—precisely what is observed empirically.

---

> ### Author Response · Authors · 2025-11-30
>
> ### Continuous-control sanity check
>
> To further verify practicality in high-dimensional continuous spaces, Figure 5-(bottom) (Hopper) reports normalized next-state discrepancies between transitions induced by $\beta$ and $\pi$, using a 3D PCA embedding as a proxy for kernel divergence. The discrepancies are modest (median ≈ 0.36, 90th percentile ≈ 0.64), and only the top 1% approach the maximal possible divergence. This again shows that most transitions lie well within half of the maximum kernel distance, providing empirical support that uniform $d$-boundedness is reasonably satisfied **in standard deep RL settings.**
>
> ## (W3, Q3) Additional experiments in discrete action space and sparse reward environments.
>
> We note that **Figure 5-(TOP) already includes a sparse-reward experiment**, CartPole-BalanceSparse, evaluated with a cap length of 100. T4 is the only model-free multi-step method that succeeds under such an extreme sparsity regime. Additional sparse-reward tasks are currently being run and will be included in the appendix.
>
> To address the reviewer’s request for discrete-action domains, we are also running experiments on **Atari environments**, and these results will likewise be added to the appendix within the review period.

---

### Official Review · Reviewer_pYMM · 2025-10-30

**Soundness:** 2
**Presentation:** 3
**Contribution:** 2
**Rating:** 4
**Confidence:** 3

**Summary:**

This paper introduces T4 (Time To Truncate Trajectory), an adaptive method for multi-step off-policy reinforcement learning that addresses the instability caused by long-horizon return. The core idea is to dynamically truncate each trajectory's return based on an estimate of the "mixing time" of the underlying policies, which is the point where the data-generating (behavior) policy and the target policy's state distributions are likely to converge. This avoids the bias and variance issues that arise from using fixed, long n-step returns.

**Strengths:**

1. It provides a new, principled way to determine the horizon for off-policy returns by linking it to Markov chain mixing times, moving beyond ad-hoc fixed horizons.
2. The T4 algorithm is proven to converge to the optimal value function, even when using arbitrary, changing behavior policies, a significant guarantee that many multi-step methods lack.
4. When integrated with state-of-the-art algorithms like SAC and TD3 on continuous control benchmarks, T4 consistently improves performance and learning speed compared to one-step and other multi-step methods.
4. The method is shown to be insensitive to the maximum truncation length hyperparameter, reducing the need for environment-specific tuning.

**Weaknesses:**

1. The convergence proofs rely on strong assumptions (like uniform ergodicity) that may not hold in all real-world environments, creating a gap between theory and practice.
2. The validation is confined to continuous control tasks (MuJoCo). Its effectiveness in other domains like discrete action spaces (e.g., Atari) or tasks with very sparse rewards has not been demonstrated.
3. The empirical analysis could be deeper. It doesn't fully explore the behavior of the truncation mechanism itself (e.g., how the chosen horizon adapts during training) or directly measure the bias-variance trade-off.
4. While robust to the main truncation cap, the method relies on a specific heuristic to estimate policy overlap, and its sensitivity to this choice or the discount factor (\gamma) is not fully explored.

**Questions:**

see weaknesses.

---

> ### Author Response · Authors · 2025-11-30
>
> We sincerely thank the reviewer for the thoughtful feedback. Several concerns appear to arise from understandable misunderstandings about our assumptions and empirical scope. To clarify these points, we have added new figures, additional analyses, and explanatory text in the revised version. We address each concern in turn.
>
> ## (W1) The convergence proofs rely on strong assumption (Definition1: uniform ergodicity).
>
>
> The reason we introduced uniform ergodicity in **Definition 1** is not to assume strong closeness between $P_\beta$ and $P_\pi$, but to formalize that stochastic truncation remains feasible
>
> **even when the disagreement parameter $d$ is large**—close to 1—so the condition becomes extremely weak.
>
> In the revision, we include a small **CliffWalking experiment (Figures 2 and 3(a))** to illustrate how Definition 1 (uniform $d$-bounded kernels)  behave in a simple but structurally challenging MDP. CliffWalking contains **absorbing states** and **irreversible transitions**, making the overlap between $P_\beta$ and $P_\pi$ particularly fragile. Even when $\pi$ and $\beta$ differ only mildly, the deterministic cliff dynamics induce
>
> - **large kernel gaps**, with
>
> $$
> d(s)=\left\| P^{\beta}(s,\cdot) - P^{\pi}(s,\cdot) \right\|_{\mathrm{TV}}\approx 1
> $$
>
> near the cliff;
>
> - **tiny overlap**, with
>
> $$
> \rho(s,s')=\sum_{x \in \mathcal{S}}\min (P^{\beta}(s,x),\;P^{\pi}(s',x) )\approx0
> $$
>
> for most state pairs, even when the policies differ only mildly. This corresponds to **the weakest regime** of Definition 1 and assumption 1.
>
> ### But meeting-time truncation still holds
>
> Figure 3(b) demonstrates that despite large $d$ and small $\rho$:
>
> - empirical $d$ rarely reaches 1 (max $\approx 0.7$);
> - $\rho$ remains small but **not zero**;
> - and, crucially, the RHS of Lemma 2 drops below 1 for all $(d,\rho)$ once $t\ge 10$.
>
> This implies
>
> $$
> \Pr(S_t^\beta = S_t^\pi) > 0 \quad \text{for all tested regimes},
> $$
>
> so the **first meeting time is finite**, and the T4 truncation remains well defined even under severe mismatch. This empirically validates the intended interpretation of Definition 1: it is **not** assuming kernels are close—it only requires that disagreement does not prevent eventual coalescence.
>
> ### **Empirical check in continuous-control (Hopper):**
>
> To further assess practicality, we embed the next-state transitions into a 3D PCA space and compute normalized L2 discrepancies (**Figure 5 bottom**). Even when $\beta$ is uniformly random:
>
> The results show:
>
> - median discrepancy ≈ 0.36
> - mean ≈ 0.37
> - 90th percentile ≈ 0.64
> - only the top 1% of samples approach the maximum possible discrepancy (=1)
>
> Thus, **typical transitions lie well below maximal mismatch**, supporting the empirical validity of the uniform $d$-bounded assumption.
>
>
> ## (W2) Missing experiments in discrete-action and sparse-reward settings
>
>
> We first note that **Figure 5-(TOP) includes the CartPole-BalanceSparse task**, a clear sparse-reward task (cap length = 100). T4 is the only model-free multi-step method that succeeds under this level of sparsity.
>
>  Additional sparse-reward experiments are currently running, and we will include the results in the appendix as soon as they are available. We are also running **Atari discrete-action experiments**, and results will be added within the review period.
>
>
> ## (W3) The behavior of bias/variance tradeoff during training
>
>
> We now include Figure 1-(b) in the revised version, which illustrates the behavior of **bias–variance tradeoff at 100K and 1M steps,** following the operator decomposition of Rowland et al. [A]. We rely on the decomposition in Proposition 2.1 of Rowland et al., which bounds the TD-error of a multi-step operator through three terms:  (1) **stochastic variance**, (2) **contraction error**, and (3) **intrinsic fixed-point bias**.
>
> $$
> \mathbb{E}\big[ | \hat{T}Q - Q^\pi |\infty \big]  \le \mathbb{E}\big[|\hat{T}Q - \tilde{T}Q|_2^2\big]^{1/2} + \Gamma| Q - \tilde{Q} |\infty+ |\tilde{Q} - Q^\pi|_2.
> $$
>
> This clarifies how multi-step operators accumulate error through:
>
> 1. stochastic variance,
> 2. contraction error,
> 3. intrinsic fixed-point bias.
>
> Color shifts from darker to lighter represent increasing cap lengths (3, 5, 10, 20).
>
> ### Key observations
>
> - **Multi-step** has small early bias but **variance explodes** at long horizons.
> - **Peng Q(λ)Q(\lambda)Q(λ)** has high early variance, slowing TD-error reduction.
> - **T4** maintains **lowest variance at all stages** while keeping bias small and stable.
>
> Thus T4 achieves the most favorable long-run bias–variance behavior.

---

> > ### Author Response · Authors · 2025-11-30
> >
> > ## (W4) The ablation on $\gamma$
> >
> > Based on Equation (2) (described as below), the term $(\gamma \lambda)^t$ appears as a coupled factor in Equation (2). Then, sweeping $\lambda$ already captures the functional dependence on $\gamma$.
> >
> > Therefore, **an ablation over $\lambda$ already effectively reflects the behavior with respect to $\gamma$.** onetheless, to fully address the concern about the final $\gamma$-term,, we will run an additional ablation and add it to the appendix within the review period.
> >
> >
> > $$
> > \mathcal{R}Q_n = Q_n + \mathbb{E}_\beta  \sum (\gamma \lambda)^t \text{...}
> > $$
> >
> >
> > ### References
> >
> > [A] Rowland, Mark, Will Dabney, and Rémi Munos. "Adaptive trade-offs in off-policy learning." *International Conference on Artificial Intelligence and Statistics*. PMLR, 2020.

---

### Official Review · Reviewer_PfVi · 2025-10-31

**Soundness:** 3
**Presentation:** 3
**Contribution:** 3
**Rating:** 4
**Confidence:** 3

**Summary:**

The authors propose T4 (Time To Truncate Trajectory), a stochastic and adaptive truncation mechanism designed to address the core difficulty of horizon selection in off-policy multi-step settings. Traditional fixed-length truncation either amplifies distribution mismatch or loses long-horizon information, leading to instability and bias. T4 resolves this by connecting truncation length to the mixing time of policy-induced Markov chains and estimating an adaptive cutoff through a coupling-based analysis. Technically, T4 reformulates the Retrace operator using Bernoulli random variables that represent stepwise disagreement between behavior and target trajectories, stochastic ally terminating credit propagation once the trajectories are expected to align. The paper establishes that T4 is non-conservative yet convergent under arbitrary behavior policies, offering both theoretical guarantees and practical robustness. Empirical results on MuJoCo and sparse-reward benchmarks demonstrate consistent gains over Retrace, Peng’s $Q(\lambda)$, and n-step baselines, with comparable sample efficiency to model-based MBPO while remaining fully model-free. Overall, the paper provides a principled integration of coupling-based theory and algorithmic design, yielding an adaptive, theoretically grounded alternative to fixed-horizon multi-step RL, though the reliance on bounded-kernel and ergodicity assumptions may restrict its direct applicability to more complex or non-stationary environments.

**Strengths:**

1. The paper proposes a principled stochastic truncation method (T4) that links horizon selection in multi-step off-policy reinforcement learning to the mixing time of the policy-induced Markov chain. This connection provides a theoretical basis for adaptive horizon control and helps address the long-standing problem of balancing bias and variance in multi-step updates.
2. The T4 operator reformulates the Retrace framework using Bernoulli random variables to stochastically determine the truncation point. It is shown to be non-conservative yet convergent under arbitrary behavior policies, offering a unified and flexible solution that avoids the instability and heavy tuning required by fixed-length multi-step methods.
3. The experiments are comprehensive and well-structured, covering standard continuous control and sparse-reward tasks. T4 consistently improves both sample efficiency and stability over baselines such as Retrace, Peng’s $Q(\lambda)$, and n-step methods, while approaching the performance of model-based algorithms like MBPO despite remaining entirely model-free.

**Weaknesses:**

1. There seems to be a semantic mismatch between the theoretical quantity $p_i = \Pr(S_i^\beta \neq S_i^\pi)$, introduced in Sec. 3 as the probability that the two coupled chains driven by $\beta$ and $\pi$ have not yet met at step $i$, and the practical estimator $\hat p_i = 1 - \min(\beta(a_i \mid s_i), \pi(a_i \mid s_i))$used in Sec. 4.1. The former is a state-/kernel-level disagreement that depends on the whole transition kernel under both policies (cf. Eq. (7)), and it is exactly this quantity that appears in the coupling argument leading to the truncated operator and the contraction statement in Thm. 1. The latter, however, is a single-sample, action-level proxy computed from the sampled pair $(s_i, a_i)$, and in general it does not control the total-variation overlap of the next-state distributions induced by $\beta$ and $\pi$. Since the theoretical guarantees hinge on the kernel-level bound, could the authors justify that replacing $p_i$ by $\hat p_i$ still yields the same contraction (or an equivalent domination inequality) required by Thm. 1? If this cannot be shown, please make explicit what the practical algorithm in Sec. 4.1 is actually guaranteed to approximate (e.g., a biased but stable truncation, or a contraction with a weaker constant).
2. It seems the experiments do not align the method in the setting it is designed for, i.e., when the behavior policy is far from the target policy or mixes slowly. The results in Sec.  5 are all on standard continuous-control tasks with SAC/TD3-like training and replay, where the behavior is close to the current policy and the off-policy gap is small, so fixed-$n$ targets and Retrace are already fairly stable. This does not show that the proposed truncation helps when $T_{\beta,\pi}$ is large or the data come from older/heterogeneous policies. Can the authors add runs with deliberately stale or mismatched behavior to demonstrate the claimed advantage?
3. The stochastic truncation intuitively introduces additional randomness on top of trajectory sampling. Please quantify its variance compared to deterministic truncation (e.g., fixed $n$) and show that the overall MSE is indeed smaller in the new scenario.

**Questions:**

Most of my concerns are already detailed in the cons section. I would be happy to raise the score if the authors can address them convincingly. One remaining question: Can the author provide a practical example when assumption 1 is satisfied?

---

> ### Author Response · Authors · 2025-11-30
>
> We sincerely appreciate the reviewer’s careful reading and the constructive feedback. In particular, we are grateful for the reviewer’s note that *“I would be happy to raise the score if the authors can address them convincingly.”* This guidance is extremely helpful, and we fully acknowledge the importance of the three concerns raised.
>
> In the revised response, we address **all three points with concrete theoretical clarifications and additional analyses.**
>
> ## (W1) the semantic gap between kernel-level $p_i$ and the practical estimator $\hat{p}_i$.
>
> Because the environment transition $P_{\text{env}}(s' \mid s,a)$ is policy-independent, the kernels $P_\beta(\cdot\mid s)$ and $P_\pi(\cdot\mid s)$ are induced by pushing $\beta(\cdot\mid s)$ and $\pi(\cdot\mid s)$ through the same stochastic map. By the data-processing inequality for total variation, we have
>
> $$
> p_i = \mathrm{TV}\big(P_\beta(\cdot\mid s_i), P_\pi(\cdot\mid s_i)\big),
> $$
>
> $$
> \tilde p_i^{\text{(act)}}  = \mathrm{TV}\big(\beta(\cdot\mid s_i), \pi(\cdot\mid s_i)\big)= 1 - \sum_a \min(\beta(a\mid s_i), \pi(a\mid s_i)),
> $$
>
> $$
> p_i \leq \tilde p_i^{\text{(act)}}.
> $$
>
> This derivation has been added as **Appendix J** in the revised version.
>
> During learning, we approximate the action-level distance using the sample-based estimator
>
> $$
>  1 -  \min(\beta(a_i\mid s_i), \pi(a_i\mid s_i)).
> $$
>
> which is a stochastic lower bound on the full action-level TV. Importantly, this approximation does **not weaken** the theoretical guarantees. Since
>
> $$
> p_i \le \tilde p_i^{(\mathrm{act})} \quad\text{and}\quad \hat p_i \le \tilde p_i^{(\mathrm{act})},
> $$
>
> the estimated meeting time can only be **larger** than the true meeting time:
>
> $$
> \hat T_{\beta,\pi} \ge T_{\beta,\pi}.
> $$
>
> Given that the truncation length satisfies
>
> $$
> K \ge \min\left((1-\gamma)^{-1},\mathbb{E}[T_{\beta,\pi}]\right),
> $$
>
> a larger disagreement estimate merely produces a **more conservative (longer)** truncation. As shown in Fig. 3-(b) (RHS of Lemma 2), this preserves the contraction regime.
>
> **Therefore, replacing $p_i$ by $\hat p_i$ still yields a valid operator, keeps contraction, and leaves all theoretical claims intact.**
>
> ## (W2, Q1) Behavior–target mismatch experiments and Assumption 1, and empirical justification
>
> We agree that substantial behavior–target mismatch is an essential regime to examine—and it is precisely where T4 is designed to be most useful.In response to the reviewer’s request, we added targeted analyses and new experiments.
>
> ---
>
> ### **1. Why Definition 1 and Assumption 1 matter — and why they are realistic**
>
> In the revision, we add a **CliffWalking** experiment analysis (Figures 2 and 3(a)) to illustrate how Definition 1 (uniform $d$-bounded kernels) and Assumption 1 (cross-Doeblin condition) behave in a simple but structurally challenging MDP. CliffWalking contains **absorbing states** and **irreversible transitions**, making the overlap between $P_\beta$ and $P_\pi$ particularly fragile. Even when $\pi$ and $\beta$ differ only mildly, the deterministic cliff dynamics induce
>
> - **large kernel gaps**, with
>
> $$
> d(s)=\left\| P^{\beta}(s,\cdot) - P^{\pi}(s,\cdot) \right\|_{\mathrm{TV}}\approx 1
> $$
>
> near the cliff;
>
> - **tiny overlap**, with
>
> $$
> \rho(s,s')=\sum_{x \in \mathcal{S}}  \min (P^{\beta}(s,x), P^{\pi}(s',x) ) \approx 0
> $$
>
> for most state pairs. This shows that **a large off-policy gap naturally yields large $d$ and small $\rho$**, even in basic tabular MDPs. In other words, the reviewer’s intuition is correct: Assumption 1 appears restrictive when the off-policy gap is large.
>
> ---
>
> ### **2. Even in hard regimes, meeting-time truncation remains meaningful**
>
> Figure 3(b) demonstrates that despite large $d$ and small $\rho$:
>
> - empirical $d$ rarely reaches 1 (max $\approx 0.7$);
> - $\rho$ remains small but **not zero**;
> - and, crucially, the RHS of Lemma 2 drops below 1 for all $(d,\rho)$ once $t\ge 10$.
>
> This implies
>
> $$
> \Pr(S_t^\beta = S_t^\pi) > 0 \quad \text{for all tested regimes},
> $$
>
> even in the presence of severe mismatch. Thus, the **first meeting time is finite**, and T4’s truncation mechanism remains well-defined and practically relevant. This empirically validates the theoretical use of Definition 1 and Assumption 1: they are not assuming that $P_\beta$ and $P_\pi$ are “close,” but rather that **their disagreement does not prevent eventual coalescence**, which is exactly the scenario demonstrated by the meeting-time behavior.

---

> ### Author Response · Authors · 2025-11-30
>
> ### **3. Empirical check in Hopper: Assumption 1 is not overly strong**
>
> To further address the reviewer’s concern about practicality in continuous spaces, we added an empirical **sanity check** in **Figure 5-bottom (Hopper)**.
>
> We embed next-state transitions from $\beta$ and $\pi$ into a 3D PCA space and measure normalized L2 discrepancies as a proxy for kernel divergence.
>
> The results show:
>
> - median discrepancy ≈ 0.36
> - mean ≈ 0.37
> - 90th percentile ≈ 0.64
> - only the top 1% of samples approach the maximum possible discrepancy (=1)
>
> Thus, **most transitions under differing policies lie within roughly half of the maximum kernel divergence**, even **when we take $\beta$ as uniformly random.** This supports the idea that the uniform $d$-boundedness and cross-Doeblin overlap are empirically satisfied in typical regions of **realistic RL environments.**
>
> Across both tabular and continuous domains:
>
> - Large mismatch → large ddd and small ρ\rhoρ (expected and observed).
> - Meeting times remain finite → T4’s truncation is always well-defined.
> - Empirical kernel discrepancies are moderate → Assumption 1 is realistic.
>
> ## (W3) Variance and MSE of stochastic truncation
>
> In Figure 1 of the revised version, we add explicit **bias–variance analyses** following the decomposition of Rowland et al. This decomposition separates multi-step TD error into: (i) stochastic variance, (ii) contraction error, and (iii) intrinsic fixed-point bias, providing a principled lens for comparing operators.
>
> ### **Empirical findings added to the revision**
>
> - **Fixed $n$-step** quickly develops **large variance** as $n$ grows, even when its early-stage bias is small.
> - **Peng($\lambda$)** exhibits **high initial variance**, limiting stable error reduction.
> - **T4** consistently shows the **lowest variance and the smallest MSE**, while keeping bias small and stable throughout training.
>
> ### **Interpretation**
>
> T4 does *not* introduce variance. Instead, by stopping credit propagation once behavior–target paths align, T4 **prevents long-horizon variance amplification**, giving a significantly better bias–variance frontier.
>
> We also clarified that “cumulative error scaling” refers to the LHS of the Rowland-style decomposition: multiplicative correction ratios, compounded bootstrap residuals, and mismatch amplification. The new experiments show that T4 mitigates exactly these effects.

---

### Official Review · Reviewer_nSE5 · 2025-11-01

**Soundness:** 2
**Presentation:** 2
**Contribution:** 3
**Rating:** 4
**Confidence:** 3

**Summary:**

This paper proposes T4 (Time To Truncate Trajectory), a stochastic and adaptive truncation mechanism for multi-step off-policy reinforcement learning. The core idea is to treat the truncation horizon as a stochastic variable determined by the meeting time between trajectories sampled under the behavior and target policies.
By linking this meeting time to the mixing time of the underlying Markov chain through coupling analysis, the paper provides a principled framework for adaptive truncation within the Retrace formulation.

**Strengths:**

1. Novel theoretical framing: The paper introduces a coupling-based view of trajectory truncation, connecting horizon length to mixing time — an elegant and original perspective that generalizes Retrace.
2. Solid theoretical guarantees: The contraction and convergence results (Theorems 1 & 2) are well-motivated and broadly applicable under mild assumptions ($p_i \le \xi$, $\gamma < \tfrac{1}{1+\xi}$).
3. Relevance: Horizon selection and cumulative variance amplification remain open issues in multi-step off-policy RL, and this work moves the discussion toward adaptive control of truncation length.

**Weaknesses:**

1. Empirical ambiguity:
- Figure 1 claims to show “cumulative error scaling,” yet it only displays average return vs. environment steps; no error metric or bias/variance quantification is presented.
- Figure 3 shows marginal improvements—SAC-T4 only occasionally underperforms SAC. Also, I would suggest changing the color of either MBPO or SAC, as they are hard to distinguish visually.
- Figure 4 is less naturally integrated: the paper switches from MuJoCo continuous control to CartPole without explaining why, how λ-values are chosen, or how sparse rewards relate to previous experiments.

2. Fairness and tuning:
In Figure 4, T4 is tested with $\lambda \in \{0.3, 0.7, 0.9, 1\}$ while Peng $Q(\lambda)$ is shown only for $\lambda \in \{0.9, 1\}$, producing an unbalanced comparison. The figure shows better performance for T4 at $\lambda = 0.3, 0.7$, but Peng $Q(\lambda)$ with the same $\lambda$ values is never tested. Besides, when sharing  the same $\lambda$ ($0.9, 1$), T4 is clearly outperformed by the Peng baseline.

3. Missing definitions:
Equation (3) introduces the total-variation (TV) norm without defining it or referring readers to Appendix A, where the definition appears later. Equation (4) uses $\Omega(\tau_{\text{mix}})$ without explaining that it is asymptotic notation (e.g., meaning “at least proportional to $\tau_{\text{mix}}$.”)

4. Figure clarity and aesthetics:
In Figure 3, like mentioned above, SAC and MBPO curves are rendered in very similar colors, making them difficult to distinguish.
Captions throughout the paper are a bit terse and omit key experimental details (e.g., environment, reward structure, hyperparameter settings).

**Questions:**

1. Could you please clarify what exactly “cumulative error scaling” refers to in Figure 1? If this corresponds to bias or variance accumulation, consider including an explicit quantitative measure (e.g., mean-squared TD error) to strengthen the connection between theory and experiment.
2. Please elaborate on the motivation for switching from MuJoCo to CartPole in Figure 4. A brief explanation of the sparse-reward setting and its relevance to your overall argument would make the comparison clearer.

Others: see Weaknesses.

---

> ### Author Response · Authors · 2025-11-30
>
> We sincerely thank the reviewer for the thoughtful and constructive feedback.
>
> We appreciate the recognition of our paper’s **coupling-based formulation**, **theoretical guarantees**, and the **importance of adaptive truncation** in multi-step off-policy RL. The comments regarding missing quantitative comparisons, $\lambda$-sweep fairness, sparse-reward justification, and clarity of definitions and figures were very helpful. All points have been addressed through additional experiments, clearer explanations, refined notation, and updated visualizations.  These changes strengthen both the technical depth and the overall presentation of the paper, and we are grateful for the reviewer’s guidance.
>
> ## **(W1,Q1) Clarification of “cumulative error scaling”**
>
> We now include Figure 1-(b), which reports **bias–variance behavior at 100K and 1M steps**, following the operator decomposition of Rowland et al. [A].
>
> This decomposition bounds the TD error into
>
> (1) stochastic variance,
>
> (2) contraction error, and
>
> (3) fixed-point bias:
>
> $$
> \mathbb{E}\big[ | \hat{T}Q - Q^\pi |\infty \big]  \le \mathbb{E}\big[|\hat{T}Q - \tilde{T}Q|_2^2\big]^{1/2} + \Gamma| Q - \tilde{Q} |\infty+ |\tilde{Q} - Q^\pi|_2.
> $$
>
> ### **Updated empirical findings**
>
> We first note that Darker to lighter colors correspond to cap lengths 3, 5, 10, and 20, respectively.
>
> - **Multi-step** begins with low bias but its **variance grows rapidly** with larger horizons, consistent with its degraded Hopper performance.
> - **Peng’s Q(λ)** exhibits **high early variance**, limiting stability despite eventual variance reduction.
> - **T4** maintains the **lowest variance across all horizons**, with a small and stable bias, yielding overall faster and more stable convergence.
>
> These results indicate that while bias decreases for all methods during training, **variance differences are structural** and persist throughout learning—explaining T4’s long-run advantage.
>
> We also clarified that **“cumulative error scaling”** refers to the LHS of Rowland’s decomposition and captures
>
> - multiplicative accumulation of per-step correction ratios,
> - compounding bootstrap residuals over long horizons, and
> - amplification of behavior–target mismatch.
>
> This matches known variance-amplification behavior in multi-step off-policy evaluation.
>
> ## **(Q2, W2) Motivation for sparse CartPole and Fairness in $\lambda$ sweeps**
>
> Thank you for highlighting this point. **CartPole-BalanceSparse is a **mujoco-based** continuous-control task from the DeepMind Control Suite**, consistent with our MuJoCo settings. We selected this task because the suite **does not provide sparse versions of Hopper, Walker, or other locomotion tasks**. CartPole-BalanceSparse is thus the official sparse benchmark available, and its sparse-reward nature makes it a natural stress-test for long-horizon truncation.
>
> To isolate the effect of traces in this setting, we followed the standard configuration and used **cap length = 100** and $\gamma= 1$.
>
> Following the reviewer’s suggestion, we added **Peng** for $\lambda$ **= 0.3 and 0.7**, matching the T4 sweep. Peng remains unstable at these λ values, even under aligned settings. The updated figure therefore provides a fully fair comparison.
>
> ## **(W3–W5) Presentation & Clarity Improvements**
>
> We appreciate the reviewer’s detailed comments. All points have been fully addressed in the revision:
>
> - The **TV norm definition** is now placed near Eq. (3) with a pointer to Appendix A.
> - **Asymptotic notation** in Eq. (4) is defined directly in the main text.
> - **Figures** have been updated with clearer color palettes, expanded captions (environment names, reward types, cap length, and λ values), and larger fonts/markers.
>
> ### References
>
> [A] Rowland, Mark, Will Dabney, and Rémi Munos. "Adaptive trade-offs in off-policy learning." *International Conference on Artificial Intelligence and Statistics*. PMLR, 2020.

---

> > ### Author Response · Authors · 2025-12-03
> >
> > We have revised the MBPO comparison figure to improve readability as requested by the reviewer.
> > Additionally, we explored a slightly adjusted hyperparameter setting for **Hopper**, which resulted in improved performance; this supplementary result and its description have been moved to the appendix due to space constraints (see Appendix, p.~19).

---

### Author Response · Authors · 2025-11-30
**Summary and Remark comments for AC and Reviewers**

Thank you for your service. We appreciate that reviewers consistently recognized the contribution of estimating a mixing-time proxy directly from sampled trajectories and using it for adaptive truncation during learning. At the same time, several reviewers shared a common concern regarding the theoretical assumptions introduced for rigor—specifically the uniform $d$-boundedness and cross-Doeblin conditions. As clarified below, these assumptions serve primarily as analytical devices, and we confirm empirically and theoretically that our method remains valid even when the conditions are extremely weak.

**For clarity, we note that all revisions in the updated manuscript are highlighted in red.**

The following summarizes the main points raised across reviews.

### **1. Mixing-time assumption & uniform ergodicity**

Reviewers noted that the assumption in *Definition 1* (uniformly d-bounded kernels / uniform ergodicity) felt strong or unclear in practical RL settings.

**Clarification added:**

- The assumption is used only to support the coupling-based upper bound.
- In practice, the condition becomes extremely weak when (d) is close to 1, and the revised draft includes a **CliffWalking** example demonstrating why this assumption is mild even in MDPs with irreversible transitions and absorbing states.
- Add empirical check on Hopper in Figure 5-(bottom)

### **2. Semantic gap between kernel-level discrepancy and practical estimator**

Multiple reviewers asked how the theoretical quantity $\mathrm{TV}(P_\beta(\cdot|s), P_\pi(\cdot|s))$ relates to the practical estimator $\hat p_i = 1 - \min(\beta(a_i|s_i), \pi(a_i|s_i))$.

**Clarification added:**

- The estimator is an *upper bound* on the TV distance due to the data-processing inequality.
- Using $\hat p_i \ge p_i$ inflates the estimated meeting time but does not break correctness; truncation length only needs to exceed $\min(E[T_{\beta,\pi}], (1-\gamma)^{-1})$.
- We add detailed explanation in Appendix J.

### **3. Need for additional quantitative bias–variance evidence**

Multiple reviewers requested explicit **bias–variance** curves beyond the main learning plots.

**Update:**

- Added Figure 1(b) showing bias–variance patterns at 100K and 1M steps, following the Rowland et al. operator decomposition.
- Clarified what “cumulative error scaling” means (multiplicative correction ratios, compounding bootstrap errors, and mismatch amplification).
- Our method, T4, has the lowest variance and the best performance during training.

### **4. Request for more experiments (sparse reward / discrete action spaces)**

A common request across reviewers:

- Evaluation on **sparse-reward tasks**,
- Evaluation on **discrete action domains** such as Atari.

    **Update:**

- We highlighted that **Figure 5-(TOP)** / CartPole-BalanceSparse is **already a sparse-reward** test where only **T4** succeeds reliably among model-free multi-step methods.
- Atari experiments are currently being executed and will be included as soon as possible.

---

> ### Author Response · Authors · 2025-12-04
> **Additional early-training diagnostics on Atari are provided in Appendix Figure 9.**
>
> To further verify the early-training behavior of our multi-step operator, we added
> an IQM comparison on **Pong** and **Breakout** at **500K agent steps** using
> **update horizon = 5** and **$\lambda = 1$** (newly included figure above).
>
> Across both environments, **C51-T4 consistently outperforms C51, Rainbow, and DQN**
> in normalized score and also exhibits **noticeably reduced variance**.
> This is particularly meaningful because 500K steps is still a low-data regime for Atari,
> where unstable multi-step returns often degrade performance.
> These results demonstrate that our stochastic truncation mechanism stabilizes
> multi-step distributional updates even in the early stages of training.
>
> As we continue running additional Atari environments with longer training budgets,
> we will report the full results in the next revision to provide a more complete picture
> of the operator’s behavior across diverse task types.

---

### Meta-Review · Area_Chair_wQuY · 2025-12-24

**Summary:**

The work identifies the horizon selection issue as the key bottleneck of multi-step off-policy RL, by arguing that the distribution mismatch between behavior and target policies can grow rapidly with longer rollouts. By analyzing the mixing time, the authors propose a stochastic truncation method called T4 to resolve this issue.

After reading the discussion between authors and reviewers, the AC finds that the authors have indeed cleared up many of the concerns. However, there are still a few questions that the AC thinks are not fully addressed or "skipped". They include: (1) The experiments are not really done for the setting it is designed for, i.e., when the behavior policy is far from the target policy or mixes slowly;  (2) the method's reliance on specific heuristics to estimate policy overlap is not fully explored; (3) lack of sufficient motivation for T4 (especially given the experiments are not what T4 aims for).

Based on the rebuttal, the AC thinks that the score of paper would  rise a bit (if the reviewers were able to adjust score like past ICLR review). However, the AC thinks this increase in score is not sufficient to achieve an accept. Therefore, the AC would like to suggest a reject to this paper.

**Reviewer Concerns:**

Addressed:
(1) Most issues raised by nSE5.
(2) PfVi: technical question on p_i.


Not Addressed:
(1) PfVi: The experiments are not really done for the setting it is designed for, i.e., when the behavior policy is far from the target policy or mixes slowly.
(2) pYMM & uLFM: strong assumption (not resolved, but sufficiently justified).
(3) uLFM: The paper fails to clarify the necessity of proposing T4 (only partially justified).
(4) PyMM: The method's reliance on specific heuristics to estimate policy overlap is not fully explored. (The authors did not reply to this comment).

**Reviewer Scores:**

The scores of nSE5 and PfVi may rise to 5 because their concerns are mostly addressed. But the others will probably not change, based on the AC's reading of response and concerns.

---

### Decision · Program_Chairs · 2026-01-26

Reject